

**Heterogeneous response of Siberian tree-ring and stable isotope proxies to the largest**
**Common Era volcanic eruptions**
Olga V. Churakova[1,2*], Marina V. Fonti[2], Matthias Saurer[3,4], Sébastien Guillet[1], Christophe
Corona[5], Patrick Fonti[3], Vladimir S. Myglan[6], Alexander V. Kirdyanov[2,7,8], Oksana V.
Naumova[6], Dmitriy V. Ovchinnikov[7], Alexander Shashkin[2,7], Irina Panyushkina[9], Ulf
Büntgen[3,8], Malcolm K. Hughes[9], Eugene A. Vaganov[2,7,10], Rolf T.W. Siegwolf[3,4], Markus
Stoffel[1,11,12]
*[1]Institute for Environmental Sciences, University of Geneva, CH-1205 Geneva, Switzerland*
*[2]Institute of Ecology and Geography, Siberian Federal University RU-660049 Krasnoyarsk,*
*Svobodniy pr 79/10, Russia*
*[3]Swiss Federal Institute for Forest, Snow and Landscape Research WSL, Zürcherstrasse 111,*
*CH-8903 Birmensdorf, Switzerland*
*[4]Paul Scherrer Institute, CH- 5232 Villigen - PSI, Switzerland*
*[5]Université Blaise Pascal, Geolab, UMR 6042 CNRS, 4 rue Ledru, F-63057 Clermont-Fer-*
*rand, France*
*[6]Institute of Humanities, Siberian Federal University RU-660049 Krasnoyarsk, Svobodniy pr*
*82, Russia*
*[7]Sukachev Institute of Forest SB RAS, Federal Research Center "Krasnoyarsk Science Cen-*
*ter SB RAS" RU-660036 Krasnoyarsk, Akademgorodok 50, bld. 28, Russia*
*[8]Department of Geography, University of Cambridge, Downing Place, Cambridge CB2 3EN*
*[9]Laboratory of Tree-Ring Research, University of Arizona, 1215 E. Lowell St., Tucson, 85721,*
*USA*
*[10]Siberian Federal University, Rectorate, RU-660049 Krasnoyarsk, Svobodniy pr 79/10, Rus-*
*sia*



[11]*dendrolab.ch, Department of Earth Sciences, University of Geneva, 13 rue des Maraîchers,*
*CH-1205 Geneva, Switzerland*
[12]*Department F.A. Forel for Aquatic and Environmental Sciences, University of Geneva, 66*
*Boulevard Carl-Vogt, CH-1205 Geneva, Switzerland*
**Corresponding author:** Olga V. Churakova*
E-Mail: olga.churakova@unige.ch



**Abstract**
Stratospheric volcanic eruptions have far-reaching impacts on global climate and society. Tree
rings can provide valuable climatic information on these impacts across different spatial and
temporal scales. Here we explore the suitability of tree-ring width (TRW), maximum latewood
density (MXD), cell wall thickness (CWT), and $\delta^{13}C$ and $\delta^{18}O$ in tree-ring cellulose for the
detection of climatic changes in northeastern Yakutia (YAK), eastern Taimyr (TAY) and Rus-
sian Altai (ALT) sites caused by six largest Common Era stratospheric volcanic eruptions (535,
540, 1257, 1640, 1815 and 1991).
Our findings suggest that TRW, MXD, and CWT show strong summer air temperature anom-
alies in 536, 541-542, 1258-1259 at all study sites. However, they do not reveal distinct and
coherent fingerprints after other eruptions. Based on $\delta^{13}C$ data, 536 was extremely humid in
YAK and TAY, whereas 541 and 542 were humid years in TAY and ALT. In contrast, the
1257 eruption of Samalas likely triggered a sequence of at least two dry summers across all
three Siberian sites.
No further extreme hydro-climatic anomalies occurred at Siberian sites in the aftermath of the
1991 eruption. Summer sunshine duration decreased significantly in 536, 541-542, 1258-1259
in YAK, and 536 in ALT. Conversely, 1991 was very sunny in YAK. Since climatic responses
to large volcanic eruptions are different, and thus affect ecosystem functioning and productivity
differently in space and time, a combined assessment of multiple tree-ring parameters is needed
to provide a more complete picture of past climate dynamics, which in turns appears funda-
mental to validate global climate models.
**Key words:** $\delta^{13}C$ and $\delta^{18}O$ in tree-ring cellulose, tree-ring width, maximum latewood den-
sity, cell wall thickness, drought, temperature, precipitation, sunshine duration, vapor pres-
sure deficit



## 1. Introduction

Stratospheric volcanic eruptions can substantially modify the Earth's radiative balance and cool the troposphere. This is due to the massive injection of sulphate aerosols which are able to reduce surface temperatures on timescales ranging from months to years (Robock, 2000). The global cooling associated with the radiative effects of volcanic aerosols, which absorb terrestrial radiation and scatter incoming solar radiation significantly, has been estimated to about 0.5°C during the two years following the Mount Pinatubo eruption in June 1991 (Hansen et al., 1996).

Since trees – as living organisms are impacted in their metabolism by environmental changes, their responses to these changes are recorded in the biomass, as it is found in tree-ring parameters (Schweingruber, 1996). The decoding of tree-ring archives therefore is used to reconstruct climate of the past. A summer cooling of the Northern Hemisphere (NH) ranging from 0.6°C to 1.3°C has been reported after the Common Era (CE) 1257 Samalas, 1452/3 Unknown, 1600 Huaynaputina, and 1815 Tambora eruptions based on tree-ring width (TRW) and maximum latewood density (MXD) reconstructions (Briffa, 1998; Schneider et al., 2015; Stoffel et al., 2015; Wilson et al., 2016; Esper et al., 2017; Guillet et al., 2017).

According to climate simulations, significant changes in the precipitation regime can also be expected after large volcanic eruptions; these include, among others, rainfall deficit in monsoon prone regions and in Southern Europe (Joseph and Zeng, 2011) as well as wetter than normal conditions in Northern Europe (Robock and Liu 1994; Gillet et al., 2004; Peng et al., 2009; Meronen et al., 2012; Iles et al., 2013; Wegmann et al., 2014). However, despite recent advances in the field, the impacts of stratospheric volcanic eruptions on the hydro-climatic variability at regional scales remain largely unknown. Therefore, this relevant knowledge about moisture anomalies is critically needed, especially at high-latitude sites where tree growth is mainly limited by summer temperatures.



As dust and aerosol particles of large volcanic eruptions affect primarily the radiation regime,
three major drivers of plant growth, i.e. photosynthetic active radiation (PaR), temperature and
vapor pressure deficit (VPD) will be affected by volcanic activity. This is reflected in reduced
TRW as a result of reduced photosynthesis but even more so by low temperature. As cell divi-
sion is strongly temperature dependent, its rate (tree-ring growth) will exponentially decrease
with decreasing temperature below 3–7°C (Körner, 2015), outweighing the "low light / low-
photosynthesis" effect by far. Furthermore, over the last years, some studies using mainly car-
bon isotopic signals ($\delta^{13}C$) in tree rings showed eco-physiological responses of trees to volcanic
eruptions at mid-latitudes (Battipaglia et al., 2007). By contrast, both carbon ($\delta^{13}C$) and oxygen
($\delta^{18}O$) isotopes in tree rings have been rarely employed to trace CE volcanic eruptions in high-
latitude (Churakova (Sidorova) et al., 2014; Gennaretti et al., 2017) or high-altitude (Sidorova
et al., 2011) proxy records.
Previous studies indicate that approaches including TRW, MXD and cell wall thickness (CWT)
as well as $\delta^{13}C$ and $\delta^{18}O$ in tree cellulose are a promising way to disentangle hydro-climatic
variability as well as winter and early spring temperatures at high-latitude and high-altitude
sites (Sidorova et al., 2008, 2010, 2011; Churakova (Sidorova) et al., 2014). In that sense, re-
cent work has allowed the retrieval of high-resolution, seasonal information on water and car-
bon limitations on growth during spring and summer from CWT measurements (Panyushkina
et al., 2003; Sidorova et al., 2011; Fonti et al., 2013; Bryukhanova et al., 2015). Depending on
site conditions, $\delta^{13}C$ variations reflect light (stand density) (Loader et al., 2013), water availa-
bility (soil properties) and air humidity (proximity to open waters, i.e. rivers, lakes, swamps
and orography) as these parameters have been recognized to modulate the stomatal conduct-
ance ($g_l$) controlling carbon isotopic discrimination.
Schematically, depending on study site, stratospheric volcanic eruptions will lead to decreased
temperatures, increased humidity and reduction of light intensity; therefore, one may expect to



observe a decrease in carbon isotope ratio due to limited photosynthetic activity and high sto-
matal conductance. By contrast, volcanic eruptions have also been credited for an increase in
photosynthesis as dust and aerosol particles cause an increased light scattering, compensating
for the light reduction (Gu et al., 2003). A significant increase in $\delta^{13}C$ values in tree-ring cel-
lulose after a volcanic event should be interpreted as an indicator of drought (stomatal closure)
or high photosynthesis. But such an enhancement after volcanic events would only occur when
temperature and humidity are not below a certain threshold.
In the past, very limited attention has been given to the elemental and isotopic composition of
tree rings in years during which they may have been subjected to the climatic influence of
powerful, but remote, tropical volcanic eruptions. Yet, a multi-proxy approach –-as outlined
above – would help to deepen our understanding of the complex climatic impacts of strato-
spheric eruptions as postulated by models at the regional scale (through a reduction in irradia-
tion, temperature and VPD, resulting in reduced TRW, $\delta^{13}C$ and $\delta^{18}O$).
In this study, we aim to fill this gap by investigating the response of different components of
the Siberian climate system (i.e. temperature, precipitations, VPD, and sunshine duration) to
the largest volcanic events of the last two millennia. By doing so, we seek to extend our under-
standing of the effects of volcanic eruptions on climate by combining multiple climate sensitive
variables measured in tree rings that were formed around the time of the largest CE eruptions.
We focus our investigation on remote, two high-latitude (northeastern Yakutia), YAK and east-
ern Taimyr (TAY) and one high-altitude (Russian Altai, ALT) Siberian sites, where long tree-
ring chronologies with high climate sensitivity exist. Therefore, we developed a dataset includ-
ing five tree-ring proxies: TRW, MXD, CWT, $\delta^{13}C$ and $\delta^{18}O$ stable isotope chronologies de-
rived from larch trees to (1) determine the major climatic drivers of the above mentioned prox-
ies and to evaluate their suitability in terms of climate responsiveness, for each proxy separately



and in combination; and (2) based on these analyses reconstruct the climatic effect of these
unusually large CE volcanic eruptions (Table 1).

**2. Material and methods**

*2.1. Study sites*
The study sites are situated in Siberia (Russian Federation), far away from industrial centers,
in zones characterized by continuous permafrost in northeastern Yakutia (YAK, 69°N, 148°E);
eastern Taimyr (TAY, 70°N, 103°E) and in the Altai mountains (ALT, 50°N, 89°E) (Fig. 1a,
Table 2). Tree-ring samples were collected during several expeditions and included old relict
wood and living larch trees, *Larix cajanderi* Mayr (max. 1216 years) in YAK, *Larix gmelinii*
Rupr. (max. 640 years) in TAY and *Larix sibirica* Ldb. (max. 950 years) in ALT. TRW chro-
nologies have been developed and published in the past (Fig. 1, Hughes et al., 1999; Sidorova
and Naurzbaev 2002; Sidorova 2003 for YAK; Naurzbaev et al., 2002 for TAY; Myglan et al.,
2008 for ALT).
Mean annual air temperature is lower at the high-latitude YAK and TAY sites than at the high-
altitude ALT site (Table 2). Annual precipitation totals are very low for all study sites. The
vegetation period calculated with a growth threshold of +5º C (Fritts 1976; Schweingruber
1996) is very short (50-120 days) at all locations (Table 2). Sunshine duration for tree growth
is higher at YAK and TAY (ca. 18-20 h/day in summer) compared to ALT (ca. 18 h/day in
summer) (Sidorova et al., 2005; Myglan et al., 2008; Sidorova et al., 2011; Churakova (Si-
dorova) et al., 2014).



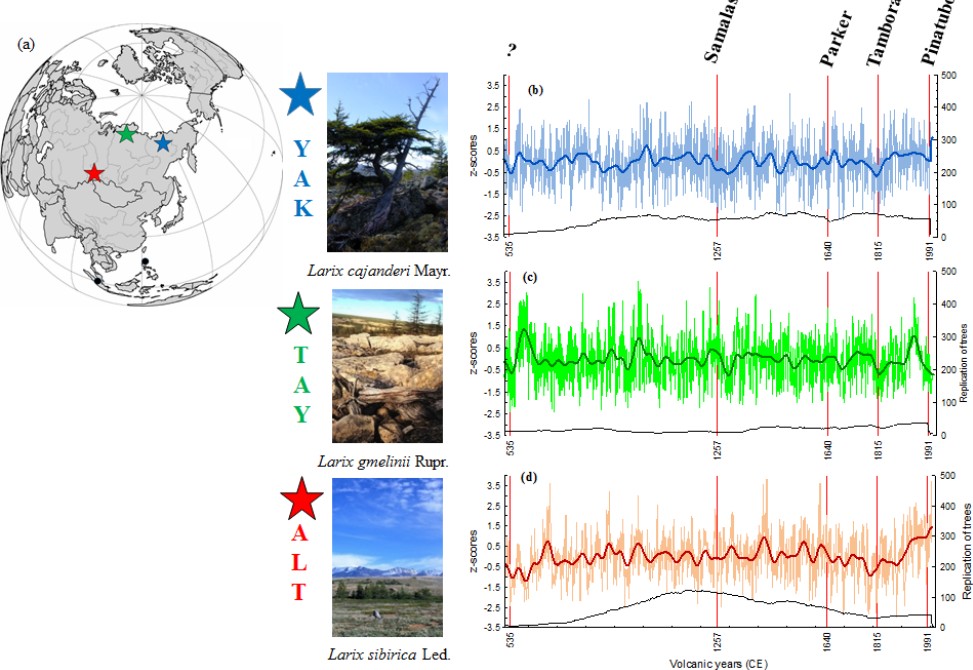

**Fig. 1.** Map with the locations of the study sites (stars) and volcanic eruptions (black circles)
considered in this study (a). Annual tree-ring width index (light lines) and smoothed by 51-
year Hamming window (bold lines) chronologies from northeastern Yakutia (YAK - **blue**, b)
(Hughes et al., 1999; Sidorova and Naurzbaev 2002; Sidorova 2003), eastern Taimyr (TAY -
**green**, c) (Naurzbaev et al., 2002), and Russian Altai (ALT - **red**, d) (Myglan et al., 2009) were
constructed based on larch trees (Photos: V. Myglan – ALT, M. M. Naurzbaev – YAK, TAY).

*2.2. Selection of the study periods and larch subsamples*
Volcanic aerosols deposited in ice core records (Gao et al., 2008; Crowley and Untermann,
2013; Sigl et al., 2015) attest to 6 major volcanic events in CE 535, 540, 1257, 1640, 1815, and
1991, that may have had a noticeable impact on the climate system globally. These events rank
as the 12th (18.81±6.94 Tg[S]), the 3rd (31.85±7.73 Tg[S]), the 1st (59.42±10.86 Tg[S]), the
13th, (18.68±4.28 Tg[S]), the 4th (28.08±4.49 Tg[S]) and 27th (9 ± 2 Tg[S]) largest volcanic



events of the last 1500 years in terms of stratospheric sulphur injection (Toohey and Sigl,

182    2017).

To investigate climatic impacts of these eruptions in Siberian regions we developed MXD,
CWT, $\delta^{13}C$ and $\delta^{18}O$ chronologies for the following periods around (± 10 years): CE 525-545,
1247-1267, 1630-1650, 1805-1825, and 1950-2000, with the latter being used to calibrate tree-
ring proxy versus available climate data (Table 2).
Material was prepared from the 2000-yr long TRW chronologies available at each of the sites
from the previous studies (Fig. 1 b-d). According to the level of conservation of the material,
the largest possible number of samples was prepared for each of the proxies. Unlike TRW,
which could be measure on virtual all samples, some of the material was not conserved well-
enough to allow for tree-ring anatomy and stable isotope analysis. At least 4 tree samples cov-
ering the volcanic periods were used for CWT and stable isotope analyses, a sample size that
is smaller as compared to TRW or MXD studies, but perfectly in line with the standards of
replication used for CWT and isotopes in reference studies (Loader et al., 1997; Panyushkina
et al., 2003; Fonti et al., 2013).





**Table 1.** List of stratospheric volcanic eruptions used in the study.

| Study period (CE) | Date of eruption Month/Day/Year | Volcano name | Volcanic Explosivity Index (VEI) | Location, coordinates | References |
|---|---|---|---|---|---|
| 525–545 | NA/NA/535 | Unknown | ? | Unknown | Stothers, 1984 |
| | NA/NA/540 | Unknown | ? | Unknown | Sigl et al., 2015 |
| 1247-1267 | May-October/NA/ 1257 | Samalas | 7 | Indonesia, 8.42°N, 116.47°E | Lavigne et al., 2013; Stothers, 2000; Sigl et al., 2015 |
| 1630-1650 | December/26/1640 | Parker | 5 | Philippines, 6°N, 124°E | Zielinski et al., 1994 |
| 1805-1825 | April/10/1815 | Tambora | 7 | Indonesia, 8°S, 118°E | Zielinski et al., 1994 |
| 1950 - 2000 | June/15/1991 | Pinatubo | 6 | Philippines, 15°N, 120°E | Zielinski et al., 1994; Sigl et al., 2015 |

NA – not available.


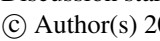



**Table 2.** Summary of tree-ring sites in northeastern Yakutia (YAK), eastern Taimyr (TAY), and Altai (ALT) and weather stations used in the
study. Monthly air temperature (T, °C), precipitation (P, mm), sunshine duration (S, h/month) and vapor pressure deficit (VPD, kPa) data were
used from the available meteorological database: http://aisori.meteo.ru/ClimateR.

| Site | Species | Location | Weather station | Meteorological parameters | | | | Length of vegetation period (day) | Thawing permafrost depth (max, cm) | Annual air temperature (°C) | Annual precipitation (mm) |
|---|---|---|---|---|---|---|---|---|---|---|---|
| | | | | T (°C) Periods | P (mm) | S (h/month) | VPD (kPa) | | | | |
| YAK | *Larix cajanderi* Mayr. | 69°N, 148°E | Chokurdach 62°N, 147°E, 61 m. a.s.l. | 1950-2000 | 1966-2000 | 1961-2000 | 1950-2000 | 50-70* | 20-50* | -14.7 | 205 |
| TAY | *Larix gmelinii* Rupr. | 70°N, 103°E | Khatanga 71°N, 102°E, 33m. a.s.l. | 1950-2000 | 1966-2000 | 1961-2000 | 1950-2000 | 90** | 40-60** | -13.2 | 269 |
| ALT | *Larix sibirica* Ledeb. | 50°N, 89°E | Mugur Aksy 50°N, 90°E 1850 m. a.s.l. | 1963-2000 | 1966-2000 | | | 90-120*** | 80-100*** | -2.7 | 153 |
| | | | Kosh-Agach 50°N, 88°E 1758 m.a.s.l. | | | 1961-2000 | 1950-2000 | | | | |

*Abaimov, 1996; Hughes et al., 1999; Churakova (Sidorova) et al., 2016
**Naurzbaev et al., 2002
***Sidorova et al., 2011



*2.3. Tree-ring width analysis*
Ring width of 12 trees was re-measured for each selected period. Cross-dating was checked by
comparison with the existing complete 2000-yr TRW chronologies (Fig. 1). The TRW series were
standardized using the ARSTAN program (Cook and Krusic, 2008) based on the negative expo-
nential curve (k>0) or a linear regression (any slope) prior to averaging with the biweight robust
mean (Cook and Kairiukstis 1990). Signal strength in regional TRW chronologies was assessed
with the Expressed Population Signal (EPS) statistics as it measures how well the finite sample
chronology compares with a theoretical population chronology based on an infinite number of
trees (Wigley et al., 1984). For each period, the EPS exceeded the cutoff point of 0.85, implying
that the estimated broad-scale environmental signal was not resulting from anomalies in the indi-
vidual series.

*2.4. Image analysis of cell wall thickness (CWT)*
Analysis of wood anatomical features was performed for all studied periods with an AxioVision
scanner (Carl Zeiss, Germany). Micro-sections were prepared using a sliding microtome and
stained with methyl blue (Furst, 1979). Tracheids in each tree ring were measured along five radial
files of cells (Munro et al., 1996; Vaganov et al., 2006) selected for their larger tangential cell
diameter (T). For each tracheid, CWT and the radial cell diameter (D) were computed. In a second
step, tracheid anatomical parameters were averaged for every tree ring. Site chronologies are pre-
sented for the complete annual ring chronology without standardization due to the absence of low-
frequency trend. CWT data from ALT for the periods 1790-1835 and 1950-2000 were used from
the past studies (Sidorova et al., 2011; Fonti et al., 2013) and for YAK for the period from 1600-
1980 from Panyushkina et al., (2003).

*2.5. Maximum latewood density (MXD)*



Maximum latewood density chronologies from ALT were available continuously for the period
CE 1407-2007 from Schneider et al., (2015) and for YAK and TAY the period CE 1790-2004
from Sidorova et al., (2010). For any of the other periods, at least six cross-sections (for CE 516-
560, only four sections could be used, as this period is less well replicated) were sawn with a
double-bladed saw, to a thickness of 1.2 mm, at right angles to the fiber direction. Samples were
exposed to X-rays for 35-60 min (Schweingruber 1996). MXD measurements were obtained with
a resolution of 0.01 mm, and brightness variations transferred into (g•cm$^3$) using a calibration
wedge (Lenz et al., 1976; Eschbach et al., 1995) from a Walesch X-ray densitometer 2003. All
MXD series were detrended in ARSTAN by calculating subtractions from straight-line functions
(Fritts, 1976). Site chronologies were developed for each volcanic period using the bi-weight ro-
bust mean.

*2.6. Theory on stable isotope fractionation ($\delta^{13}C$ and $\delta^{18}O$)*
During photosynthetic $CO_2$ assimilation $^{13}CO_2$ is discriminated against $^{12}CO_2$, leaving the newly
produced assimilates depleted in $^{13}C$. The carbon isotope discrimination ($^{13}\Delta$) is partitioned in the
diffusional component with a = 4.4‰ and the biochemical fractionation with b = 27‰, for C3
plants, during carboxylation via Rubisco. The $^{13}\Delta$ is directly proportional to the $c_i/c_a$ ratio, where
$c_i$ is the leaf intercellular, and $c_a$ the ambient $CO_2$ concentration. This ratio reflects the balance
between stomatal conductance ($g_l$) and photosynthetic rate ($A_N$). A decrease in $g_l$ at a given $A_N$
results in a decrease of $^{13}\Delta$, as $c_i/c_a$ decreases and vice versa. The same is true when $A_N$ increases
or decreases at a given $g_l$. Since $CO_2$ and $H_2O$ gas exchange are strongly interlinked with the C-
isotope fractionation $^{13}\Delta$ is controlled by the same environmental variables i.e. PaR, $CO_2$, VPD
and temperature (Farquhar et al., 1982, 1989; Cernusak et al., 2013).
The oxygen isotopic compositions of tree-ring cellulose record the $\delta^{18}O$ of the source water de-
rived from precipitation, which itself is related to temperature variations at middle and high lati-
tudes (Craig, 1961; Daansgard, 1964). It is modulated by evaporation at the soil surface and to a



larger degree by evaporative and diffusion processes in leaves; the process is largely controlled by
the vapor pressure deficit (Dongmann et al., 1972, Farquhar and Loyd, 1993, Cernusak et al.,
2016). A further step of fractionation occurs as sugar molecules are transferred to the locations of
growth (Roden et al., 2000). During the formation of organic compounds the biosynthetic frac-
tionation leads to a positive shift of the $\delta^{18}O$ values by 27‰ relative to the leaf water (Sternberg,
2009). The oxygen isotope variation in tree-ring cellulose therefore reflects a mixed climate infor-
mation, often dominated by a temperature, source water or sunshine duration modulated by the
VPD influence.

*2.7. Stable isotope analysis in tree cellulose ($\delta^{13}C$ and $\delta^{18}O$)*
The cross-sections of relict wood and cores from living trees used for the TRW, MXD and CWT
measurements were then selected for the isotope analyses. We analysed four subsamples for each
studied period according to the standards and criteria described in Loader et al., (2013). The first
50 yrs. of each sample were excluded to limit juvenile effects (McCarroll and Loader, 2004). After
splitting annual rings with a scalpel, the whole wood samples were enclosed in filter bags. α-
cellulose extraction was performed according to the method described by Boettger et al., 2007.
For the analyses of $^{13}C/^{12}C$ and $^{18}O/^{16}O$ isotope ratios, 0.2-0.3 mg and 0.5-0.6 mg of cellulose were
weighed for each annual ring, into tin and silver capsules, respectively. Carbon and oxygen iso-
topic ratios in cellulose were determined with an isotope ratio mass spectrometer (Delta-S, Finni-
gan MAT, Bremen, Germany) linked to two elemental analyzers (EA-1108, and EA-1110 Carlo
Erba, Italy) via a variable open split interface (CONFLO-II, Finnigan MAT, Bremen, Germany).
The $^{13}C/^{12}C$ ratio was determined separately by combustion under oxygen excess at a reactor tem-
perature of 1020°C. Samples for $^{18}O/^{16}O$ ratio measurements were pyrolyzed to CO at 1080°C
(Saurer et al., 1998). The instrument was operated in the continuous flow mode for both, the C and
O isotopes. The isotopic values were expressed in the delta notation relative to the international
standards (Eq. 1):



$$\delta \text{ sample} = R_{sample}/R_{standard}\text{-}1 \qquad \text{(Eq. 1)}$$
where $R_{sample}$ is the molar fraction of $^{13}C/^{12}C$ or $^{18}O/^{16}O$ ratio of the sample and $R_{standard}$ the molar
fraction of the standards, Vienna Pee Dee Belemnite (VPDB) for carbon and Vienna Standard
Mean Ocean Water (VSMOW) for oxygen. The precision is $\sigma \pm 0.1‰$ for carbon and $\sigma \pm 0.2‰$
for oxygen. To remove the atmospheric $\delta^{13}C$ trend after CE 1800 from the carbon isotope values
in tree rings (i.e. Suess effect, due to fossil fuel combustion) we used atmospheric $\delta^{13}C$ data from
Francey et al., (1999), http://www.cmdl.noaa.gov./info/ftpdata.html). These corrected series were
used for all statistical analyses. The $\delta^{18}O$ cellulose series were not detrended.

*2.8. Climatic data*
Meteorological series were obtained from local weather stations close to the study sites and used
for the computation of correlation functions between tree-ring proxies and monthly climatic pa-
rameters (Table 2). Sunshine duration data were obtained from available Kosh-Agach meteoro-
logical station (http://aisori.meteo.ru/ClimateR).

*2.9. Statistical analysis*
All chronologies for each period were normalized to z-scores (Fig. 2). To assess post-volcanic
climate variability, we used Superposed Epoch Analysis (SEA, Panofsky and Brier, 1958) with
the five proxy chronologies available at each of the three study sites. In this experiment, the 15
years before and after a volcanic eruption were analyzed. SEA is applied to the six annually dated
volcanic eruptions (Table 1).
To test the sensitivity of the studied tree-ring parameters to climate, bootstrap correlation functions
have been computed between proxy chronologies and monthly climate predictors using the
'bootRes' package of R software (R Core Team 2016) for the period 1950 (1966)-2000.
To estimate whether volcanic years can be considered as extreme, we computed Probability Den-
sity Functions (PDFs, Stirzaker, 2003) for each study site and for each tree-ring parameter over a



period of 221 years for which measurements are available (Fig. S1). A year is considered (very)
extreme if the value of a given parameter is below the (5$^{th}$) 10$^{th}$ percentile of the PDF.

**3. Results**

*3.1. Anomalies in tree-ring proxy chronologies after stratospheric volcanic eruptions*
According to the SEA, decreased values have frequently been observed in the proxy chronologies
over one or two years after the volcanic eruptions. For instance, the TRW chronologies show neg-
ative deviations the year following the eruptions at YAK and ALT with significant anomalies in
CE 536 (-2.7 σ and -1.8 σ for YAK and ALT respectively) and a delayed decrease, two years after
the events, at TAY (Fig. 2). Comparable, although less pronounced patterns of variation are rec-
orded in the MXD chronologies with decreasing values for ALT (-4.4 σ) and YAK (-2.8 σ) in CE
537, and even less pronounced patterns of variation for TAY (Fig. 2). In this regard, the sharpest
decrease is observed in the CWT chronologies from YAK (-3.9σ), TAY (-3.0 σ), and ALT (- 2.9
σ), one and two years after the eruptions, respectively (Fig. 2). The δ$^{18}$O chronologies show a
distinct decrease one year after the eruptions for YAK -3.9 σ, in the year of 1259, TAY -3.0σ in
537, and ALT - 2.9 σ in 537 only (Fig. 2, Fig. S1). Finally, δ$^{13}$C negative anomalies are observed
in TAY, and – to a lesser extent – in YAK two years after almost all of the eruptions, but are
largely absent from the ALT chronology. The CE 540 eruption was recorded in CWT and δ$^{13}$C at
YAK site only (Fig. 2).
Overall, the SEA shows the high spatiotemporal variability and complexity of the response of the
Siberian climate system to the largest volcanic events of the CE. The eruption of CE 535 induced
extremely narrow tree rings at the three sites associated with extremely low MXD values in YAK
and ALT. A lagged response by one year is observed in the CWT proxies at all three sites.





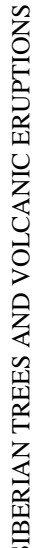

**Fig. 2.** Normalized (z-score) individual tree-ring index chronologies (TRW, **black**), cell wall thickness (CWT, **green**), δ¹³C (**red**) and δ¹⁸O (**blue**) in tree-ring cellulose chronologies from YAK, TAY and ALT for the specific periods 15 years before and after the eruptions CE 535, 1257, 1640, 1815 and 1991 are presented. Vertical lines showed year of the eruption.



The behavior of isotope chronologies is more complex, with a distinct decrease in $\delta^{13}C$ at the
high-latitude sites (YAK, TAY), whereas $\delta^{18}O$ series are impacted only at the high-altitude
ALT site.
With respect to the CE 1257 Samalas eruption (Fig. 2), the year following the eruption was
recorded as very extreme in the TRW and CWT chronologies at all sites whereas very extreme
anomalies were recorded in $\delta^{13}C$ for CE 1259 (see Fig. S1). The impacts of the more recent CE
1640 Parker, 1815 Tambora, and 1991 Pinatubo eruptions are, by contrast, by far less obvious.
In CE 1643, extreme decreases are observed in the TRW and CWT series of the high-latitude
sites YAK and TAY, whereas tree-ring proxies are not clearly affected at ALT. No extreme
anomalies are observed in CE 1816 in Siberia regardless of the site and the tree-ring parameter
analyzed. The ALT $\delta^{13}C$ chronology can be seen as an exception to the rule here as it evidenced
extreme values in CE 1817. Finally, the Pinatubo eruption is captured in CE 1992 by MXD and
CWT chronologies from YAK and classified as extreme in the CWT and $\delta^{18}O$ chronologies
from ALT in 1993 (Fig. S1, right panel).

*3.2. Tree-ring proxies versus meteorological series*

*3.2.1. Monthly air temperatures and sunshine duration*
Bootstrapped functions evidence significant positive correlations (*p<0.05*) between TRW and
MXD chronologies and mean summer (June-July) temperatures at all sites. Temperatures at the
beginning (June) and the end of the growing season (mid-August) influenced the MXD chro-
nology in ALT (r = 0.57) and YAK (r = 0.55), respectively (Fig. 3). July temperatures appear
as a key factor for determining tree growth as they significantly impact CWT, $\delta^{13}C$, and $\delta^{18}O$
(with the exception of TAY for the latter) chronologies (r=0.28-0.60) at YAK and ALT.




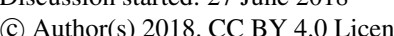

**Fig. 3.** Significant correlation coefficients between tree-ring parameters: TRW, MXD, CWT,

$\delta^{13}$C and $\delta^{18}$O versus weather station data: temperature (**red**), precipitation (**blue**), vapor pres-

sure deficit (**green**), and sunshine duration (**yellow**) from September of the previous year to

August of the current year for three study sites were calculated. Table 2 lists stations used in

the analysis.

Namely, February and March temperatures affected significantly $\delta^{18}$O as recorded in the cellu-

lose chronologies at YAK, ALT (r=0.25, r=0.26), while March and May (r=0.30) temperatures

in TAY, respectively.



Correlation analysis between July temperature and July sunshine duration showed significant
correlation for YAK (r=0.56) and ALT (r=0.34). July sunshine duration are strongly and posi-
tively correlated with $\delta^{18}O$ in larch tree-ring cellulose chronologies from YAK (r=0.73) and
ALT (r=0.51) for the period 1961-2000.

*3.2.2. Monthly precipitation*
The strongest July precipitation signal is observed at ALT (r=-0.54) and TAY (r=-0.51) with
$\delta^{13}C$ chronologies (*p<0.05*). In addition, at ALT a positive relationship is observed between
March precipitation and TRW (*p<0.05*) (r=0.37), MXD (r=0.32) and CWT (r=0.34), respec-
tively. At YAK, July precipitation showed negative relationship with $\delta^{18}O$ in tree-ring cellulose
(r=-0.34; *p<0.05*) only.

*3.2.3. Vapor pressure deficit (VPD)*
June VPD is significantly and positively correlated with the $\delta^{18}O$ chronology from ALT (r=0.67
*p<0.05*, respectively) for the period 1950-2000. The $\delta^{13}C$ in tree-ring cellulose from YAK cor-
relate with July VPD only (r=0.69 *p<0.05*). We did not find a significant influence of VPD in
TAY tree-ring and stable isotope parameters.

*3.2.4. Synthesis of the climate data analysis*
In summary, we found that during the instrumental period of weather station observations (Ta-
ble 2) mainly summer temperature influenced TRW, MXD and CWT from the HL sites (YAK,
TAY), while stable carbon and oxygen isotopes were affected by summer precipitation (YAK,
TAY, ALT), sunshine duration (YAK, ALT), and vapor pressure deficit (YAK, ALT) signals.

*3.3. Response of Siberian larch trees to climatic changes after the major volcanic eruptions*



Based on the statistical analysis above for the calibration period, we assumed that these rela-
tionships would not change over time and will provide information about climatic changes dur-
ing past volcanic periods (Fig. 4).



**Fig. 4.** Response of larch trees from Siberia to the CE volcanic eruptions (Table 1) with per-

centile of distribution considered as very extreme (< 5th, intensive color), extreme (>5th, <10th,



light color) and non-extreme (>10th, white color). July temperature changes presented as a
square from **heavy blue** (cold) to **light blue** (moderate). Summer vapor pressure deficit (VPD)
variabilities shown as a circle from **purple** (low), light purple (moderate decrease) to **orange**
(increase, developing to dry air). July precipitation presented as a rhomb from **heavy turquoise**
(wet), **light blue** (moderate) to **orange** (dry). Low July sunshine duration shown as black tri-
angle, while high – as yellow.

*3.3.1. Temperature proxies*
We found strong summer air temperature anomalies at all sites after the 535 and 1257 CE vol-
canic eruptions. The temperature decrease was found in the TRW and CWT datasets at all sites,
and also in the MXD datasets at YAK and ALT (Fig. 4). For the volcanic eruptions in later
centuries, the evidence for a decrease in temperature was not as pronounced. Namely, no strong
drop in summer temperature was found for ALT in CE 1642 nor 1643, an extreme cold in TAY
for 1643 only, while still a cold summer in YAK for both years; 1816 was cold only in YAK
(based on the CWT chronology), but not at the other sites. CE 1992 was recorded as a cold one
based on MXD and CWT from YAK, but again not for the other sites; CE 1993 was an extreme
year for ALT based on CWT and $\delta^{18}$O.

*3.3.2. Moisture proxies: precipitation and VPD*
Based on the climatological analysis with the local weather stations data (Table 2, Fig. 3) for
all studied sites we considered $\delta^{13}$C in cellulose chronologies as proxies for precipitation
changes. Yet, CWT from ALT could be considered as a proxy with mixed temperature and
precipitation signal (Fig. 3, Fig. 4). Therefore, CE 536 was extremely humid in YAK and TAY,
as well as 541 and 542 in TAY and ALT. CE 1258 was dry in YAK and ALT, while drier than
normal conditions occurred in 1259 for all studied sites. CE 1641 was dry in TAY; 1642 in



YAK and ALT. A rather wet summer was in TAY during 1815 and 1816 years. CE 1991 was
wet in YAK, 1992 in ALT followed by a dry summer in 1993 (Fig. 4).

*3.3.3. Sunshine duration proxies*
Instrumental measurements of sunshine duration (Table 2) in YAK and ALT during the recent
period showed a significant link with $\delta^{18}O$ cellulose. Based on this we conclude that sunshine
duration decreased significantly in 536, 541, 542, 1258 and 1259 in YAK, and 536 in ALT.
Conversely, summer 1991 in YAK was very sunny (Fig. 4).

**4. Discussion**
Here we present periods with large volcanic eruptions of the CE using long-term tree-ring
multi-proxy chronologies for $\delta^{13}C$ and $\delta^{18}O$, TRW, MXD, CWT for the high-latitude (YAK,
TAY) and high-altitude (ALT) sites. The main goal was to explore the suitability of the above-
mentioned proxies for the detection of abrupt climatic changes caused by volcanic eruptions:
(i) for each proxy alone, and (ii) for the combined use of all proxies, to reconstruct the respective
climatic changes, which should go beyond temperature. Since trees as living organisms respond
to various climatic impacts, the carbon assimilation and growth patterns accordingly leave
unique "finger prints" in the photosynthates, which is recorded in the wood of the tree rings
specifically and individually for each proxy.

*4.1. Evaluation of the applied proxies in Siberian tree ring data*
This study clearly shows that each proxy has to be analyzed and interpreted specifically for its
validity on each studied site and evaluated for its suitability for the reconstruction of abrupt
climatic changes.



*4.1.1. TRW, CWT and MXD*
TRW in temperature-limited environments is a strong proxy for temperature reconstructions,
as growth is a temperature-controlled process. Temperature clearly determines the duration of
the growing season and the rate of cell division (Cuny et al., 2014). With decreasing tempera-
ture, the time needed for cell division increases exponentially, particularly in a threshold range
between 3–7°C (Körner, 2015). Accordingly, low growing season temperatures are reflected in
narrow tree rings. The upper temperature limit is species and biome specific. In most cases tree
growth is limited by drought rather than by high temperatures, since water shortage and VPD
increase with increasing temperature. Still this does not make TRW a suitable proxy to deter-
mine the influence of water availability and air humidity, especially at the temperature-limited
sites.
MXD chronologies obtained for the Eurasian subarctic record mainly a July-August tempera-
ture signal (Vaganov et al., 1999; Sidorova et al., 2010; Büntgen et al., 2016) and add valuable
information about climate conditions toward the end of the growth season. Similarly, CWT is
an anatomical parameter, which contains information on carbon sink limitation of the cambium
due to extreme cold conditions (Panyushkina et al., 2003; Fonti et al., 2013; Bryukhanova et
al., 2015). The clear signal about reduced number of cells within a season, for example, strong
decreasing CWT in CE 536 at YAK or formation of frost rings in ALT (CE 536-538, 1259) has
been shown in our study.

*4.1.2. Stable carbon isotope ratio*
The carbon isotope ratio ($^{13}C/^{12}C$), a useful proxy for water availability, air humidity and PaR
is impacted by these variables through their effects on $A_N$ and $g_l$. Indirectly it indicates low
temperatures as under these conditions VPD is low with high $g_l$ values and moderate $A_N$, result-
ing in more negative $\delta^{13}C$ values (high $c_i/c_a$ ratio). Furthermore, a reduction in photosynthesis



caused by volcanic dust veils is also reflected in low $\delta^{13}C$ values. For the distinction whether
$\delta^{13}C$ is predominantly determined by $A_N$ or $g_l$ the combined evaluation with $\delta^{18}O$ or TRW is
needed.

*4.1.3. Oxygen isotope ratio*
The oxygen isotope ratio ($^{18}O/^{16}O$) a widely accepted temperature proxy is varied at two levels:
First, when water vapor condenses, precipitation will carry a distinct O-isotope signature related
to condensation temperature (Daansgaard, 1964). Rained out precipitation water with its tem-
perature specific $^{18}O$ signature falls through the atmosphere and mostly falls through canopies,
which enhance evaporation and fractionation before it infiltrates into the soil. There, its varia-
tion is dampened before the water is absorbed by the roots (Sprenger et al., 2017). In non-
limiting soil water conditions no isotopic fractionation is observed during root water uptake and
its transport to the leaves (Dawson et al., 2004, Vargas et al., 2017). Second, water that enters
the leaves is subject to evaporative enrichment due to transpiration (Cernusak et al., 2016). This
process is driven by i) an analogue of relative humidity (the ratio of the partial water vapor
pressure of the ambient air ($e_a$) versus that of the leaf intercellular spaces ($e_i$), (Dongmann et
al., 1972), ii) the back diffusion of water vapor from the atmosphere into the leaf (Lehmann et
al., 2018) and iii) $g_s$, which controls transpiration (Péclet effect) (Farquhar and Lloyd, 1993).
In this study, the variation in $\delta^{18}O$ reflected the changes in temperature and sunshine duration
fairly well. It is important to consider that sunshine duration is an indirect proxy for the leaf
temperature signal. The stronger the irradiation the higher the heating effect leading to an en-
hanced evaporation in the leaves (Beerling et al., 1994). As a result $H_2^{18}O$ is more enriched in
the leaf water and $\delta^{18}O$ in organic matter is higher. As VPD is positively correlated with $\delta^{18}O$,
we can conclude that high $\delta^{18}O$ values indicate high VPD, which VPD induces a reduction in
stomatal conductance, reducing the back diffusion of depleted water molecules from ambient





air. This confirms a sunny year CE 1991 in YAK and to some extent in ALT with warm and
dry weather conditions. Interestingly, we also find less negative values for $\delta^{13}$C in the same
period. This shows that the two isotopes correlate with each other and this indicates the need
for a combined evaluation of the C and O isotopes (Scheidegger et al., 2000) taking into account
the suggested precautions (Roden and Siegwolf, 2012).

*4.2. Lag between volcanic events and response in tree rings*
In most of the discussed events, we observe a certain delay – or lag – between the eruption and
the response in tree rings of one year or more. This lag is explained by the tree's use of stored
carbohydrates, which are the substrate for leaf/needle and early wood production. These stored
carbohydrates carry the isotopic signal of previous years and depending on their remobilization
and use mask the signals in freshly produced biomass. The signal seems to be stronger if the
impacts of the eruption (i.e. dust veil, dimming) arrive to the study site late in the growth period.

*4.3. Temperature and sunshine duration changes after stratospheric volcanic eruptions*
Correlation functions show that MXD and CWT (with the exception of TAY in the latter case),
and to a lesser extent also TRW chronologies, portray the strongest signals for summer (June-
August) temperatures. In addition, significant information about sunshine duration can be de-
rived from the YAK and ALT $\delta^{18}$O series. Thus, we hypothesize that extremely narrow TRW
and very negative anomalies observed in the MXD and CWT chronologies of YAK and to a
lesser extent at ALT, in CE 536 and 1258 along with low $\delta^{18}$O values (except for ALT in CE
1257) reflect cold conditions in summer. Presumably, the temperatures were below the thresh-
old values (Körner, 2015). This hypothesis of a generalized regional cooling after both erup-
tions is further confirmed by the occurrence of frost rings at all sites in CE 536 (Myglan et al.,
2008; Guillet et al., 2017), as well as in neighboring Mongolia (D'Arrigo et al., 2001). The



unusual cooling in CE 536 is also evidenced by a very small number of cells formed at YAK
(Churakova (Sidorova) et al., 2014). According to the CWT chronologies, this cooling likely
persisted throughout the region in CE 537 and was limited to TAY and ALT in CE 538 with
formation of frost rings in ALT. Although $\delta^{18}O$ is an indirect proxy for needle temperature, low
$\delta^{18}O$ values in CE 536 and 1258 for YAK and ALT are a result of low irradiation, leading to
low temperature and low VPD (high stomatal conductance), both likely a result from volcanic
dust veils.
Similarly, in the aftermath of the Samalas eruption, the persistence of summer cooling is limited
to CE 1259 only at the three study sites, which is in line with findings of Guillet et al., (2017).
Interestingly, a slight decrease in oxygen isotope chronologies – which can be related to low
levels of summer sunshine duration (i.e. low leaf temperatures) – allows for hypothesizing that
cool conditions could have prevailed.
For all later high-magnitude CE eruptions, temperature-sensitive tree-ring proxies do not evi-
dence a generalized drop in summer temperatures. Paradoxically, the impacts of the Tambora
eruption, known for its triggering of a widespread "year without summer" (Harrington, 1992),
did only induce abnormal CWT at YAK, but no anomalies are observed at sites TAY and ALT,
except for the positive deviation of $\delta^{13}C$ (Fig. 2). While these findings may seem surprising,
they are in line with the TRW and MXD reconstructions of Briffa et al., (1998) or Guillet et al.,
(2017), who found contrasting impacts of the CE 1815 Tambora event in Eastern Siberia and
Alaska using TRW and MXD data only. The inclusion of CWT chronologies, not used in their
reconstructions, further confirm the absence of a significant cooling in this region following the
second largest eruption of the last millennium.
Finally, in CE 1992, our results evidence cold conditions in YAK, which is consistent with
weather observations showing that the below-average anomalies in summer temperatures (after
Pinatubo eruption) were indeed limited to Northeastern Siberia (Robock, 2000). In contrast,





inferences about sunny conditions in CE 1991 in YAK – and to some extent in ALT – are
confirmed by higher $\delta^{13}C$ (warm and dry) and $\delta^{18}O$ (sunny and dry) values, both indicating
warm and dry conditions. As both isotopes indicate a reduction in stomatal conductance, we
can conclude that warm (in agreement with MXD and CWT) and dry conditions were prevalent
for YAK and ALT at this time. This isotopic constellation was confirmed by the positive rela-
tionships between VPD and $\delta^{18}O$ and $\delta^{13}C$ for YAK and ALT.
However, temperature and sunshine duration are not always highly coherent over time due to
the influence of other factors, like Arctic Oscillations as it was suggested for Fennoscandia
regions by Loader et al., (2013).

*4.4. Moisture changes*
Water availability is a key parameter for Siberian trees as they are growing under extremely
continental conditions with hot summers and cold winters, and even more so with very low
annual precipitation (Table 2). Continuous permafrost, in addition, is playing a crucial role, and
can be considered as a buffer for additional water sources during hot summers (Sugimoto et al.,
2002; Boike et al., 2013; Saurer et al., 2016). Yet, thawed permafrost water is not always avail-
able for roots due to the surficial structure of the root plate or extremely cold water temperature
(close to 0ºC), which can hardly be utilized by trees (Churakova (Sidorova) et al., 2016). Thus,
Siberian trees are highly susceptible to drought, induced by dry and warm air during July and
therefore the stable carbon isotopes can be sensitive indicators of such conditions. After vol-
canic eruptions, however, low light intensity due to dust veils induce low temperatures and
reduced VPD, the driver for evapotranspiration. Under such conditions drought stress is un-
likely to occur. However, the transition phases with changes from cool and moist to warm and
dry conditions are more critical when drought is more likely to occur.





In our study, higher $\delta^{13}$C values in tree-ring cellulose indicate increasing drought conditions as
a consequence of reduced precipitation for two years after the CE 1257 volcanic eruption at all
three sites. A local drought developed at YAK at the beginning of CE 1643, while a shift to
dryer conditions was observed at TAY in the beginning of summer CE 1815 until 1820. No
further extreme hydro-climatic anomalies occurred at Siberian sites in the aftermath of the
Pinatubo eruption.

*4.5. Synthetized interpretation from the multi-parameter tree-ring proxies*
Our analysis demonstrates the added value of a tree-ring derived multi-proxy approach to better
capture the climatic variability after large volcanic eruptions. Besides the well-documented ef-
fects of temperature derived from TRW and MXD, CWT, stable carbon and oxygen isotopes
in tree ring cellulose provide important and complementary information about moisture and
sunshine duration changes (an indirect proxy for leaf temperature effective for air-to-leaf VPD)
after stratospheric volcanic eruptions.
In detail, our results reveal a complex behavior of the Siberian climatic system to the largest
eruptions of the Common Era. The CE 535 and CE 1257 Samalas eruptions caused substantial
cooling – very likely induced by dust veils (Churakova (Sidorova) et al., 2014; Guillet et al.,
2017; Helama et al., 2018) – as well as humid conditions at the high-latitude sites. Conversely,
only local and frequently delayed climate responses were observed after the CE 1641 Parker,
1815 Tambora, and 1991 Pinatubo eruptions. Similar site-dependent impacts were found in CE
1453, 1458 and 1601 (Fig. S1), frequently referred to as the coldest summers of the last millen-
nium in the Northern Hemisphere based on TRW and MXD reconstructions (Schneider et al.,
2015; Stoffel et al., 2015; Wilson et al., 2016; Guillet et al., 2017). This absence of widespread
and intense cooling or reduction of precipitation over vast regions of Siberia may result from
the location and strength of the volcanic eruption, atmospheric transmissivity as well as from



the modulation of radiative forcing effects by regional climate variability. These results are
consistent with other regional studies, which interpreted the spatio-temporal heterogeneity of
tree responses to past volcanic events (Esper et al., 2017) in terms of regional climate peculiar-
ities.

**5. Conclusions**
In this study, we demonstrate that the consequences of volcanic eruptions on climate are com-
plex and heterogeneous between sites and among events. That said, we also show that each
proxy alone can not provide the full information on an eruption but that it contributes to the
understanding and the full picture by adding to a single, specific factor, which is critical for a
comprehensive description of climate dynamics induced by volcanism and the inclusion of
these phenomena in global climate models. Therefore, the application of a multiple tree-ring
parameter approach provides much more detailed information. The multi-proxy approach al-
lows refining the interpretation and improves our understanding of the heterogeneity of climatic
signals after CE stratospheric volcanic eruptions, which are recorded in multiple tree-ring and
stable isotope parameters from the vast Siberian regions.

**Author contribution:** TRW analysis was performed at V.N. Sukachev Institute of Forest SB
RAS by O.V. Churakova (Sidorova), D.V. Ovchinnikov, V.S. Myglan and O.V. Naumova.
CWT analysis was carried out at the V. N. Sukachev Institute of Forest SB RAS, Krasnoyarsk,
Russia by M. Fonti and at the University of Arizona by I. Panyushkina. Stable isotope analysis
was conducted at the Paul Scherrer Institute (PSI), by O. V. Churakova (Sidorova), M. Saurer,
and R. Siegwolf. MXD measurements were realized with a DENDRO Walesh 2003 densitom-
eter at WSL and at the V.N. Sukachev Institute of Forest SB RAS, Krasnoyarsk, Russia by O.
V. Churakova (Sidorova) and A. V. Kirdyanov. Samples from YAK and TAY were collected




by M. M. Naurzbaev. All authors contributed significantly to the data analysis and paper writ-
ing.
**Acknowledgements:** This work was supported by Marie Curie International Incoming Fellow-
ship [EU_ISOTREC 235122], Re-Integration Marie Curie Fellowship [909122] and UFZ
scholarship [2006], RFBR [09-05-98015_r_sibir_a] granted to Olga V. Churakova-Sidorova;
SNSF M. Saurer [200021_121838/1]; Era.Net RUSPlus project granted to M. Stoffel [SNF
IZRPZ0_164735] and RFBR [№ 16-55-76012 Era_a] granted to E.A. Vaganov; project granted
to Vladimir S. Myglan RNF, Russian Scientific Fond [№ 15-14-30011]; Alexander V. Kirdya-
nov was supported by the Ministry of Education and Science of the Russian Federation
[#5.3508.2017/4] and RSF [#14-14-00295]; Scientific School [3297.2014.4] granted to Eugene
A. Vaganov; and US National Science Foundation (NSF) grants [#9413327, #970966,
#0308525] to Malcolm K. Hughes and US CRDF grant # RC1-279, to Malcolm K. Hughes
and Eugene A. Vaganov. We thank Tatjana Boetgger for her support and access to the stable
isotope facilities within UFZ Haale/Saale scholarship 2006; Anne Verstege, Daniel Nievergelt
for their help with sample preparation for the MXD and Paolo Cherubini for providing lab
access at the Swiss Federal Institute for Forest, Snow and Landscape Research (WSL).






**Figure legend**

**Fig. 1.** Map with the locations of the study sites (stars) and volcanic eruptions (black circles)
considered in this study (a). Annual tree-ring width index (light lines) and smoothed by 51-year
Hamming window (bold lines) chronologies from northeastern Yakutia (YAK - **blue**, b)
(Hughes *et al.,* 1999; Sidorova 2003), eastern Taimyr (TAY - **green**, c) (Naurzbaev *et al.,*
2002), and Russian Altai (ALT - **red**, d) (Myglan *et al.,* 2009) were constructed based on larch
trees (Photos: V. Myglan – ALT, M. M. Naurzbaev – YAK, TAY).

**Fig. 2.** Normalized (z-score) individual tree-ring index chronologies (TRW, **grey**), maximum
latewood density (MXD, **black**), cell wall thickness (CWT, **green**), $\delta^{13}$C (**red**) and $\delta^{18}$O (**blue**)
in tree-ring cellulose chronologies from YAK, TAY and ALT for the specific periods 15 years
before and after the eruptions CE 535, 1257, 1640, 1815 and 1991 are presented. Vertical lines
showed year of the eruption.

**Fig. 3.** Significant correlation coefficients between tree-ring parameters and weather station
data: temperature (**red**), precipitation (**blue**), vapor pressure deficit (**green**), and sunshine du-
ration (yellow) from September of the previous year to August of the current year for three
study sites were calculated. Table 2 lists stations used in the analysis.

**Fig. 4.** Response of larch trees from Siberia to the CE volcanic eruptions (Table 1) with per-
centile of distribution considered as very extreme (< 5th, intensive color), extreme (>5th, <10th,
light color) and non-extreme (>10th, white color). July temperature changes presented as a
square from **heavy blue** (cold) to **light blue** (moderate). Summer vapor pressure deficit (VPD)
variabilities shown as a circle from **purple** (low), **light purple** (moderate decrease) to **orange**



(increase, developing to dry air). July precipitation presented as a rhomb from **heavy turquoise**
(wet), **light blue** (moderate) to **orange** (dry). Low July sunshine duration shown as black tri-
angle, while high – as yellow.

**Table 1.** List of stratospheric volcanic eruptions used in the study.

**Table 2.** Summary of tree-ring sites in northeastern Yakutia (YAK), eastern Taimyr (TAY) and
Altai (ALT), and weather stations used in the study. Monthly air temperature (T, °C), precipi-
tation (P, mm), sunshine duration (S, h/month) and vapor pressure deficit (VPD, kPa) data were
used from available meteorological database http://aisori.meteo.ru/ClimateR.




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
