# Peer review of "Siberian tree-ring and stable isotope proxies as indicators of temperature and moisture"

_Climate of the Past, 2018_

## Referee Comment (RC1) · Anonymous Referee #1 · 27 Jul 2018

Review of Churakova et al. "Heterogeneous response of Siberian tree-ring and stable isotope proxies to the largest Common Era volcanic eruptions" for Climate of the Past Discussions

This study takes measurements of multiple tree-ring parameters (including stable isotopes and cell wall thickness) for larch chronologies at 3 sites in N and S Siberia. The measurements are mostly focussed on 20-year periods centred on 6 major volcanic eruptions, as well as during the post-1950 period for comparison with instrumental climate observations. The experimental design is good as it makes sense to first focus these measurements (some of which are expensive to do in both time and money) on

periods where we might a priori expect a strong climatic signal to reveal interesting insights into the multi-variate response across tree ring variables and infer multi-variate changes in climate (temperature, precipitaiton, humidity, sunshine). The findings could then help guide future resources to the sites and parameters that would add most value, as well as learning about the complex responses to volcanic erutpions that likely go beyond the summertime cooling that has already been established.

However, there are problems with the data analysis and presentation of results which severely limit the value of the paper as it currently stands.

1. The lack of superposed epoch analysis.

No superposed epoch analysis is presented (despite the mention of it on line 305) to establish a best estimate of the "typical" response to a large eruption, instead the volanic epochs are only considered individually. The statistical testing on the individual events is therefore of limited statistical power because the size of the volcanic signal has to be equivalent to the 5 (or 10) percentile of the estimated variability of the time-series for it to have even a 50% chance of being identified as significant. [Also, Fig. S1 where these tests are reported is illegible at the size and resolution provided – I'm surprised this wasn't picked up by the journal technical staff for correction prior to starting the review process.]

It is good to see the responses to individual events – and I like the overall design of Fig. 4 for this purpose – but it is also necessary to see the composite behaviour because of the additional statistical power that compositing (superposing) the events will bring and the different statistical testing that would then be applied. While the purpose may be to illustrate the varied behaviour after each event, it is first necessary to see the aggregate behaviour. Once this is established, the heterogeneity can be considered.

2. There is no discrimination between (i) different responses to each event and (ii) "random" sampling variability that will make each case appear different anyway.

If the purpose is indeed to demonstrate the heterogeneous responses between events (instead of, or in addition to, the differences between sites and between tree-ring/climate parameters) then the analysis needs to consider how to discriminate truly different responses from sampling variability. Different values will occur due to internal climate/weather variability as well as error variance in the data. The statistical tests compare each value to the timeseries variability to see if they are significantly different from zero – but what is needed is to see if they are significantly different from either each other or from the composite mean (see point 1). That would demonstrate the heterogeneity between events is real and not just down to sampling variablility.

3. Errors and limitations in the presentation and discussion of the results in Fig. 4.

The results are hard to follow and their description/discussion is not presented concisely. In part the structure leads to duplication, e.g. the process-based description of stable isotopes is split between 2.6 before the results and 4.1.2/3 after the results but the results don't really confirm or alter our understanding of these processes so it seems unecessary to split this into the two sections (that would be appropriate if it was to say what our understanding was before this study and then to present the updated understanding after it, but that isn't really the case here). I wonder if the sample sizes are too small for the CWT and isotopes to really be confident in the findings – can anything further be done to show whether the sample sizes are adequate? Are there enough samples to calculate RBAR and EPS as a measure of the common signal and chronology confidence?

These limitations are compounded by errors in the description of the results in Fig. 4. Here are lines 439-449 with my comments in [CAPS]:

———— "Therefore, CE 536 was extremely humid in YAK and TAY, as well as 541 and 542 [NOTHING SIGNIFICANT IN FIG 4 IN 542] in TAY and ALT. CE 1258 was dry in YAK and ALT, while drier than normal conditions occurred in 1259 for all studied sites. CE 1641 [1641 ISN'T EVEN IN FIG 4] was dry in TAY; 1642 in YAK [NO, 1643!] and ALT

[NOT ACCORDING TO FIG 4]. A rather wet summer was in TAY during 1815 [THIS YEAR IS NOT IN THE FIGURE!] and 1816 years [NOTHING SIGNIFICANT HERE]. CE 1991 [NOT IN THE FIGURE!] was wet in YAK, 1992 in ALT [NO ALT DOESN'T APPEAR WET IN 1995 IN FIG 4] followed by a dry summer in 1993 [NOT ACCORDING TO FIG 4!] (Fig. 4).

3.3.3. Sunshine duration proxies Instrumental measurements of sunshine duration (Table 2) in YAK and ALT during the recent period showed a significant link with $\delta18O$ cellulose. Based on this we conclude that sunshine duration decreased significantly in 536, 541, 542, 1258 and 1259 in YAK, and 536 in ALT. Conversely, summer 1991 in YAK was very sunny [NOT ACCORDING TO FIG 4! BUT 1993 in ALT WAS, WHY NOT MENTION THIS?] (Fig. 4)." ————

For Pinatubo, the instrumental data could be used to confirm the multi-variate explanations, e.g. do they confirm the statements you give about 1991-1993 on lines 442-443, 449 and 517?

Some description of Fig. 3 looks incorrect too, e.g. identification of months with significant correlations:

L370: "with the exception of TAY for the latter" - and CWT too? L381: "May" - looks like "April"? L391: "March" - looks like "April" for CWT?

Specific comments and questions:

L58: "triggered by" implies a causal link, which hasn't been established by the analysis here.

L85: Briffa –> Briffa et al.

L120: Why do you expect increased humidity?

L146: Why 6 eruptions and not more to increase the sample size? Why these particular 6? L180-181 says these are not the top 6, why not choose the top 6 in terms of

stratospheric sulphur injection?

L190: virtual –> virtually

L191, 238-239: give the actual sample size for each site, period and eruption in a table (perhaps adding to table 1) rather than just "at least 4"

L193: "perfectly" isn't needed

L229: did you consider using CWT averaged only over the latewood, so it is closer to MXD?

L304-305: It doesn't look like you subtracted the mean prior to the eruption – this is standard for SEA.

L306, 345: 15 years? Only 10 are shown in fig 2

L325: -4.4 sigma is NOT less pronounced than -1.8 sigma

L327: -3.9 sigma CWT for YAK is not visible in fig 2, which eruption do you mean?

L334: no this is not the "response" to the volcanoes. This misconception is a repeated weakness of the manuscript: the complexity show may be due to a fairly stable response to the forcing, but with local climate variability and errors in the chronologies (sample size is small for some parameters) superimposed to give individual realisations that differ from case to case.

L340: Fig. 2 has a gap in the CWT series for TAY in 536. Explain why and what this means. Frost rings?

L362 onwards: Climate analysis comments:

You only analyse correlations between individual months but later you suggest Jun-Aug temperature response for some parameters. Showing correlations for a 2 or 3 month seasonal mean might be useful, it might give a stronger correlation because intra-seasoal variability would be reduced.

Did you detrend the climate data before calculating correlations?

You need to consider cross-correlations between the climate variables, since this could explain why some tree-ring parameters are correlated with multiple climate variables. e.g. L389, could the negative correlations with MJJ precip arise because precip in these months is negatively correlated with temperature at ALT?

L372: Fig. 3 axis labelling is too small and blurry.

L410: CWT label is missing for YAK.

L468: "strong" seems, err, too strong, when correlations are only 0.3 to 0.5. Also make clear it is SUMMER temperature.

L483-484: First time that the NUMBER of cells is mentioned, similarly for frost rings.

L509: what is "gs"?

L510: at TWO sites

L514: positive correlation with VPD is signficant at only ONE site

L525: or explained by a delayed/sustained climate response or by the aerosol forcing persisting and perhaps taking some time to reach the highest latitudes?

---

## Referee Comment (RC2) · Anonymous Referee #2 · 29 Aug 2018

This manuscript attempts to investigate the response of multiple tree-ring proxies, such as ring width (TRW), maximum latewood density (MXD), cell wall thickness (CWT), and $\delta13C$ and $\delta18O$ in tree-ring cellulose derived from northeastern Yakutia (YAK), eastern Taimyr (TAY) and Russian Altai (ALT) sites, to six major volcanic events, and further to explore climate change (temperature, precipitations, VPD, and sunshine duration) caused by these six major stratospheric volcanic eruptions during the past 1500 years. They argued that heterogeneous response of Siberian tree-ring and stable isotope proxies to these major volcanic eruptions occurred. The heterogeneous response varied with complexity in terms of timing of volcanic eruptions, sites and tree-ring proxy types.

[Figure]

The results are not strange because only three sites with various climate types and different and longitudes latitudes and different tree-ring proxies representing different climatic signals are chosen for this study. Generally, distinct temperature-sensitive tree-ring records from a large number of tree-ring sampling sites across the northern Hemisphere are chosen for the detection of climate-volcanic relationship. In this case, it can maximize common climate signals (especially temperature signal) represented by tree-ring records and minimize heterogeneous local/regional internally generated variability, and consequently allowing the effect of external forcing factors such as volcanic events to be addressed. Different approach was adopted in this study. The authors chose five tree-ring proxies from three sites with different longitudes and latitudes and elevations and focus on six time segments of volcanic eruption events during the past 1500 years. It is easy to be expected that heterogeneous climate response to the volcanic eruptions represented by Siberian tree-ring and stable isotope proxies would occur because different climate or synoptic and circulation conditions are characterized by the three studied sites. At any rate, some sentences would be necessary and very helpful for clarifying the differences between this topic and prior large-scale studies. In addition, advantages and uncertainties of this study related to limited three sites and multi-proxies should be also complemented.

Comments:

The title needs a change since the chosen six volcanic events do not represent the largest volcanic eruptions, as stated in lines 179-180.

The term Common Era seems to be inconsistent with the studied span 'the past 1500 years'.

Lines 72, 92, 119, 181 and Section 3.1 line: I am not sure about the wording 'Stratospheric volcanic eruptions'.

Lines 99-101 and lines 470-472 seem to be contradictory. Please check it.

[Figure]

Lines 126-127 are not clear and need rephrasing.

Fig. 1: five volcanic eruptions (vertical lines) are indicated but one eruption is missing.

From the map, the two eruption sites (two black circles) are located in tropical areas. It would be more clear to point out in the text and abstract.

Line 184: the authors stated each studies segment is 'around $\pm$ 10 years', but Fig.2 caption says 'the specific periods 15 years before and after the eruptions'. It is confusing.

Sections 2.6 and 2.7 can be combined together.

Superposed Epoch Analysis (SEA) results need significance tests to enhance scientific rigor of the relevant descriptions (for example, section 3.1).

Section 3.1 and section 3.2 should be swapped.

Fig. 2: the gray line is not clear. It is suggested to change the line color to enhance visibility.

For TRW and MXD, they are affected by both temperature and precipitation (see Fig. 3). It is difficult to separate temperature signals alone. However, the authors chose them as only temperature indicator.

Lag between volcanic events and response in tree rings is easy to understand for carbon isotope and ring width. But for oxygen and MXD, it is generally accepted that there is no legacy effect from previous year.

The authors argued for findings of the heterogeneity with different volcanic eruptions each but a potential associated mechanism is missing, for example, climate response to 1815 Tambora eruptions.

If available, additional evidence such as historical documents and long instrumental observations are much needed to strengthen the results.

---

## Author Comment (AC1) · 6 Oct 2018

Referee 1: No superposed epoch analysis is presented (despite the mention of it on line 305) to establish a best estimate of the "typical" response to a large eruption, instead the volcanic epochs are only considered individually. The statistical testing on the individual events is therefore of limited statistical power because the size of the volcanic signal has to be equivalent to the 5 (or 10) percentile of the estimated variability of the time series for it to have even a 50

Answer: We are thankful to Referee 1 for this valuable comment. In fact, we used superposed epoch analysis for our proxies and presented the results in the supplementary material (Fig. S1) in the form of probability density functions. These pdfs included all proxies for all periods (n=221) studied after the major volcanic eruptions. In the revised version, we provide new and larger figures with increased quality for better visualization. In Fig. 2, we present only 10 years before and 10 years after the event to increase the readability, even more so as we illustrate 5 different tree-ring proxies (d18O, d13C, CWT, TRW, and MXD). However, and to take the referee's suggestion into account, we performed additional analyses, displayed in Fig. 3 for the d18O (a), d13C (b), CWT (c), TRW (d) and MXD (e) chronologies. In the new version, data is displayed separately by combining the volcanic eruptions in CE 535, 1257, 1641, 1815 and 1991. Please find the new Fig. 3 below and in the revised manuscript Fig. 3 (see P. 18-19, L. 360-369). We show that trees growing at YAK responded mainly during the first year after the eruptions, whereas a two years delay occurs at TAY and ALT (P. 3, L. 61-62). Referee 1: There is no discrimination between (i) different responses to each event and (ii) "random" sampling variability that will make each case appear different anyway. If the purpose is indeed to demonstrate the heterogeneous responses between events (instead of, or in addition to, the differences between sites and between tree-ring/climate parameters) then the analysis needs to consider how to discriminate truly different responses from sampling variability. Different values will occur due to internal climate/weather variability as well as error variance in the data. The statistical tests compare each value to the time series variability to see if they are significantly different from zero – but what is needed is to see if they are significantly different from either each other or from the composite mean (see point 1). That would demonstrate the heterogeneity between events is real and not just down to sampling variability.

Answer: We aimed at showing differences in the climatic signal captured after stratospheric volcanic eruptions by the d18O, d13C, CWT, TRW and MXD chronologies separately, and show that stark differences exist between sites. We applied unpaired t-test statistics to check for significance between each proxy and site. We find significant differences (p=0.014, df=40, n=21) between averaged d13C chronologies of the YAK and ALT sites. We now provide this information in the revised version of our manuscript (P.

18, L. 357-359). Referee 1: Errors and limitations in the presentation and discussion of the results in Fig. 4. The results are hard to follow and their description/discussion is not presented concisely. In part the structure leads to duplication, e.g. the process-based description of stable isotopes is split between 2.6 before the results and 4.1.2/3 after the results but the results don't really confirm or alter our understanding of these processes so it seems unnecessary to split this into the two sections (that would be appropriate if it was to say what our understanding was before this study and then to present the updated understanding after it, but that isn't really the case here). I wonder if the sample sizes are too small for the CWT and isotopes to really be confident in the findings – can anything further be done to show whether the sample sizes are adequate? Are there enough samples to calculate RBAR and EPS as a measure of the common signal and chronology confidence?

Answer: Thank you for this very valuable feedback. We revised the discussion and removed unnecessary repetitions in sections 4.1.2 and 4.1.3 parts. In the revised version, we restrict the discussion to a single 4.1 chapter. We have calculated RBAR and EPS values of stable isotope chronologies for the period 1950–2000, for which individual trees were analyzed separately, and show that a common signal indeed exists with an EPS>0.85. In previous studies, like Sidorova et al. (2008 JGR Biogeosciences; 2010 Global Change Biology; Climatic Change), pooled material was used for this analysis. For all other tree-ring parameters, EPS also exceeds the threshold of 0.85 (P.12, L. 212-215).

Referee 1: L58: "triggered by" implies a causal link, which hasn't been established by the analysis here.

Answer: we removed "triggered by" and replaced it by "..led to.."

Referee 1: L85: Briffa –> Briffa et al.

Answer: corrected

Referee 1: L120: Why do you expect increased humidity?

Answer: We expect a decrease in the carbon isotope ratio as a result of limited photosynthetic activity and higher stomatal conductance, which results from higher moisture conditions (i.e. increased relative humidity). To clarify our statement, we changed the sentence as follows: "Depending on the study site, a decrease in the carbon isotope ratio can be expected after stratospheric volcanic eruptions due to limited photosynthetic activity and higher stomatal conductance, which in turn would be the result of decreased temperatures, VPD and a reduction in light intensity (P. 6, L. 120-123). Referee 1: L. 146 Why 6 eruptions and not more to increase the sample size? Why these particular 6? L180-181 says these are not the top 6, why not choose the top 6 in terms of stratospheric sulphur injection?

Answer: In this study we focused only on very large volcanic eruptions with a Volcanic Explosivity Index (VEI) exceeding 5 (P. 6, L. 136 in the revised version; see Table 1). Eruptions exceeding this threshold are considered to be strong enough to affect tree growth over large areas. However, we fully agree that future work could include weaker eruptions as well, which was beyond the goals of the present work.

Referee 1: L190: virtual –> virtually

Answer: corrected Referee 1: L191, 238-239: give the actual sample size for each site, period and eruption in a table (perhaps adding to table 1) rather than just "at least 4"

Answer: We agree that the sentence in the original version of the ms was not very clear. We changed it as follows "Unlike TRW, which could be measured on virtually all samples, some of the material was not available with sufficient quality to allow for tree-ring anatomy and stable isotope analysis. We therefore use a smaller sample size for CWT (n=4) and stable isotopes (n=4) than for TRW (n=12) or MXD (n=12). Nonetheless, replications are still comparable with those used in reference papers in the fields of CWT and isotope analyses (Loader et al., 1997; Panyushkina et al., 2003; Fonti et al., 2013) (P. 9, L. 183-189).

Referee 1: L193: "perfectly" isn't needed

Answer: removed

Referee 1: L229: did you consider using CWT averaged only over the latewood, so it is closer to MXD?

Answer: In our work, we used a common annual resolution for all tree-ring parameters. In line with this decision and in order to keep the approach uniform, we used cell-wall thickness values that were averaged per ring for each of the volcanic eruptions considered. We agree that the idea to use latewood CWT is very interesting, and might be applied later for intra-annual studies of climate-growth relationship. However, in this study we did not consider this parameter.

Referee 1: L304-305: It doesn't look like you subtracted the mean prior to the eruption – this is standard for SEA. Answer: A standardization procedure (normalizing according to the mean value) was applied for each proxy and each period separately.

Referee 1: L306, 345: 15 years? Only 10 are shown in fig 2 Answer: To increase the readability and clarity Fig. 2 displays only 10 years before and after each event (see P. 9, L. 178, P. 18 L. 342, revised version) – the text has been corrected accordingly.

Referee 1: L325: -4.4 sigma is NOT less pronounced than -1.8 sigma

Answer: We corrected the sentence as follows "Decreasing MXD values for ALT (-4.4 ïĄş) and YAK (-2.8 ïĄş) were observed in CE 537. However, for TAY, we found less pronounced patterns of variation (Fig. 2)." (P. 16, L. 321-322).

Referee 1: L327: -3.9 sigma CWT for YAK is not visible in fig 2, which eruption do you mean?

Answer: We clarified that this refers to CE 541.

Referee 1: L334: no this is not the "response" to the volcanoes. This misconception is a repeated weakness of the manuscript: the complexity show may be due to

a fairly stable response to the forcing, but with local climate variability and errors in the chronologies (sample size is small for some parameters) superimposed to give individual realisations that differ from case to case.

Answer: We believe that events are different from one another. The climatic effects of the different stratospheric volcanic eruptions reached the high latitudes represented by the chronologies illustrated here at different times within the year and varying magnitudes in terms of impacts. As a consequence, tree's response to these climatic effects are apparently quite different and also contain some local effects. Many studies based on tree-ring width or maximum latewood density only report about the cooling effect induced by volcanic eruptions. Our work, by contrast, shows that volcanic eruptions may induce changes in precipitation, vapor pressure deficit and sunshine duration. Such differences can be obtained from the same trees but by using a unique multi-proxy approach.

Referee 1: L340: Fig. 2 has a gap in the CWT series for TAY in 536. Explain why and what this means. Frost rings?

Answer: The remaining sample discs from TAY, which were not used for the stable isotope analyses, were broken when shipped back to the laboratory for anatomical analyses (V.N. Sukachev Institute of Forest, Krasnoyarsk, Russia) Thus, it was impossible to produce a clear image of the CE 536 ring from this material. As a result, wall thickness information is missing for TAY. Only one sample was left but we do not present it here for reasons of consistency and sample depth. "Unfortunately, the remaining YAK sample size was too small for anatomical analyses. Thus, it was impossible to produce a clear anatomical signal for the CE 536 ring from existing material. As a result, cell wall thickness is missing for this year at TAY (Fig. 2)." (P. 12-13, L. 231-234).

Referee 1: L362 onwards: Climate analysis comments: You only analyse correlations between individual months but later you suggest Jun- Aug temperature response for some parameters. Showing correlations for a 2 or 3 month seasonal mean might be

useful, it might give a stronger correlation because intra-seasonal variability would be reduced. Did you detrend the climate data before calculating correlations? You need to consider cross-correlations between the climate variables, since this could explain why some tree-ring parameters are correlated with multiple climate variables. e.g. L389, could the negative correlations with MJJ precip arise because precip in these months is negatively correlated with temperature at ALT?

Answer: The seasonal mean in temperature response shows significant statistical relationships with tree-ring width and maximum latewood density for YAK (June-July r=0.43; 0.45); TAY (July-August, r=0.57; 0.59; June-August r=0.44; 048); and ALT (June-July r=0.51; 0.54; June-August r=0.43; 0.56), respectively for the period from 1950 to 2004 only. Averaged seasonal temperature against stable isotope data and CWT does not show higher correlations. Meteorological data are available for a relatively short period (1966-2004 for precipitation and 1950-2000 for temperature) only and visual trends do not exist for these periods. Therefore, we did not detrend climate data prior to statistical analyses. The CRU data cannot be used for our purpose due to a lack of representation back in time. No significant correlations exist between precipitation and temperature datasets for ALT.

Referee 1: L372: Fig. 3 axis labelling is too small and blurry.

Answer: Thank you for this hint. The axis labeling of Fig. 3 (Fig. 4 in revised version, P. 21) has been enlarged and the quality of the illustration has been improved in the revised version of the manuscript.

Referee 1: L410: CWT label is missing for YAK.

Answer: corrected

Referee 1: L468: "strong" seems, err, too strong, when correlations are only 0.3 to 0.5. Also make clear it is SUMMER temperature.

Answer: We clarified "..is a proxy for summer temperature reconstructions" (P. 26, L.

476). Referee 1: L483-484: First time that the NUMBER of cells is mentioned, similarly for frost rings.

Answer: We now provide a description in the section of results as: "We observe a strong decrease in CWT in CE 536 at YAK where only two layers of cells were formed in CE 536 (as compared to an average 11-20 layers of cells)." (P.16, L. 318-319)."..formation of frost rings in ALT (CE 536-538, 1259) has been shown in our study". (P. 25, L. 489-490).

Referee 1: L509: what is "gs"?

Answer: This is gl (stomatal conductance). We removed this section to avoid repetition according to the Reviewers advice.

Referee 1: L510: at TWO sites

Answer: added

Referee 1: L514: positive correlation with VPD is significant at only ONE site

Answer: added

Referee 1: L525: or explained by a delayed/sustained climate response or by the aerosol forcing persisting and perhaps taking some time to reach the highest latitudes?

Answer: We modified sentence as follow "The delayed signal could also reflect the time needed for the dust veil to be transported to the study sites" (P. 26, L. 506-507).

Please also note the supplement to this comment:
https://www.clim-past-discuss.net/cp-2018-70/cp-2018-70-AC1-supplement.pdf

**Fig. 1.** Map with the locations of the study sites (stars) and volcanic eruptions from the tropics (black dots) considered in this study (a). Annual tree-ring width index (light lines) and smoothed by 51-year Hamming window (bold lines) chronologies from northeastern Yakutia (YAK - **blue**, b) (Hughes et al., 1999; Sidorova and Naurzbaev 2002; Sidorova 2003), eastern Taimyr (TAY - **green**, c) (Naurzbaev et al., 2002), and Russian Altai (ALT - **red**, d) (Myglan et al., 2009) were constructed based on larch trees (Photos: V. Myglan – ALT, M. M. Naurzbaev – YAK, TAY).

**Fig. 1.**

[Figure]

**Fig. 2.** Normalized (z-score) individual tree-ring index chronologies (TRWi, **pink**), maximum latewood density (MXD, **black**), cell wall thickness (CWT, **green**), $\delta^{13}C$ (**red**) and $\delta^{18}O$ (**blue**) in tree-ring cellulose chronologies from YAK, TAY and ALT for the specific periods 15 years before and after the eruptions CE 535, 1257, 1640, 1815 and 1991 are presented. Vertical lines showed year of the eruptions.

**Fig. 2.**

[Figure]

**Fig. 3.** Superposed epoch analysis of $\delta^{18}O$ (a), $\delta^{13}C$ (b), CWT (c), TRW (d) and MXD (e) chronologies for each study site and for the major volcanic eruptions in CE 535, 1257, 1641, 1815 and 1991.

**Fig. 3.**

[Figure]

**Fig. 4.** Significant correlation coefficients between tree-ring parameters: TRW, MXD, CWT, $\delta^{13}$C
and $\delta^{18}$O versus weather station data: temperature (T, red), precipitation (P, blue), vapor pressure
deficit (VPD, green), and sunshine duration (S, yellow) from September of the previous year to
August of the current year for three study sites were calculated. Table 2 lists stations used in the
analysis.

**Fig. 4.**

[Figure]

**Fig. 5.** Response of larch trees from Siberia to the CE volcanic eruptions (Table 1) with percentile of distribution considered as very extreme (< 5th, intensive color), extreme (>5th, <10th, light color) and non-extreme (>10th, white color). July temperature changes presented as a square from **heavy blue** (cold) to **light blue** (moderate). Summer vapor pressure deficit (VPD) variabilities are

**Fig. 5.**

---

## Author Comment (AC2) · 6 Oct 2018

We are thankful to Referee 2 for the advices to improve the manuscript and considered them as follows: Referee 2: The title needs a change since the chosen six volcanic events do not represent the largest volcanic eruptions, as stated in lines 179-180.

Answer: We modified the title of our manuscript as follows: "Siberian tree-ring and stable isotope proxies as indicators of temperature and moisture changes after major stratospheric volcanic eruptions". The six volcanic events selected in this study are the largest eruptions over the past 1,500 years.

[Figure]

Referee 2: The term Common Era seems to be inconsistent with the studied span 'the past 1500 years'.

Answer: We replaced to "the strongest eruptions of the past 1,500 years: CE.." (P. 4, L. 83).

Referee 2: Lines 72, 92, 119, 181 and Section 3.1 line: I am not sure about the wording 'Stratospheric volcanic eruptions'.

Answer: Not all volcanic eruptions can be classified as stratospheric. However, in our case, we considered only stratospheric volcanic eruptions; therefore, we highlighted this point in more detail in our manuscript.

Referee 2: Lines 99-101 and lines 470-472 seem to be contradictory. Please check it.

Answer: We clarified these sentences in lines 103-112 and removed the sentence that was in lines 470-472 in the original submission.

Referee 2: Lines 126-127 are not clear and need rephrasing.

Answer: To avoid confusion and misunderstanding we left this sentence away.

Referee 2: Fig. 1: five volcanic eruptions (vertical lines) are indicated but one eruption is missing.

Answer: We added a reference to the event in 540.

Referee 2: From the map, the two eruption sites (two black circles) are located in tropical areas. It would be more clear to point out in the text and abstract.

Answer: We included information about two eruption sites in the figure legend (P.8, line 165-166) and added as requested this info in the text and abstract (P. 3, line 52-53). More information about volcanic eruptions can be found in Table 1.

Referee 2: Line 184: the authors stated each studies segment is 'around $\pm$ 10 years', but Fig.2 caption says 'the specific periods 15 years before and after the eruptions'. It is confusing.

Answer: We corrected the text to 10 years (P. 17, Fig. 2, L. 339-341, legend).

Referee 2: Sections 2.6 and 2.7 can be combined together. Answer: Based on the reviewer's suggestion we combined both section 2.6 and section 2.7, and modified the subsection as following "2.6. Stable carbon ($\delta$13C) and oxygen ($\delta$18O) isotopes in tree-ring cellulose". (P. 13, L. 244).

Referee 2: Superposed Epoch Analysis (SEA) results need significance tests to enhance scientific rigor of the relevant descriptions (for example, section 3.1).

Answer: We applied unpaired t-test statistics to check significance between each proxy and each site and provided new (Fig. 3, P.18-19, L. 361-370).

Referee 2: Section 3.1 and section 3.2 should be swapped.

Answer: We disagree on this point, as, in our opinion, data should be presented first, followed by statistical relationships with climatic parameters. Therefore, we preferred to keep this order as it was originally.

Referee 2: Fig. 2: the gray line is not clear. It is suggested to change the line color to enhance visibility.

Answer: That's a relevant point. We have accordingly replaced grey color with pink, to enhance visibility, as suggested (P. 17, revised Fig.2).

Referee 2: For TRW and MXD, they are affected by both temperature and precipitation (see Fig. 3). It is difficult to separate temperature signals alone. However, the authors chose them as only temperature indicator.

Answer: Absolutely, it is difficult to separate temperature from precipitation signals in temperature-limited environments. We attempted to do so by presenting the most meaningful relations (Fig. 4, P. 20, L. 382-386) with precipitation. However, significant correlation with precipitation are clearly lacking during the summer for TRW and MXD, for example, for YAK.

Referee 2: Lag between volcanic events and response in tree rings is easy to understand for carbon isotope and ring width. But for oxygen and MXD, it is generally accepted that there is no legacy effect from previous year. The authors argued for findings of the heterogeneity with different volcanic eruptions each but a potential associated mechanism is missing, for example, climate response to 1815 Tambora eruptions.

Answer: The d18O chronology displays lag effects after volcanic events that might be due to permafrost availability and mixed source water (atmospheric precipitation and thawed permafrost water) recorded in tree-ring cellulose. This signal can be stored in melted permafrost water and captured back based on climate conditions (cooling, warming anomalies). Another reason could be that volcanic eruptions cause changes in tree physiology (e.g. damage to roots because they are not well supplied by fresh assimilates). This could affect not only growth and carbon isotopes in the following years, but also oxygen isotopes, because they also depend on root conditions and physiological properties (leaf water enrichment depends on transpiration rate and such signal is reflected partially in tree-ring cellulose). Since MXD represents late-summer climate conditions we can expect that the Tambora eruption, which took place only in April 1815 would not be visible in the 1815 ring. In a study by Esper et al. (2017) based on 20 Northern Hemispheric MXD chronologies, a strong and coherent post-volcano signal was seen, mainly in the year after the eruption. Esper, J., Büntgen, U., Hartl-Meier, C., Oppenheimer, C., Schneider, L.: Northern Hemisphere temperature anomalies during 1450s period of ambiguous volcanic forcing. Bull. Volcanology. 79, 41, 2017. An absence of widespread and intense cooling or reduction of precipitation over vast regions of Siberia over the past half millennia may result from the location and strength of the volcanic eruption, atmospheric transmissivity as well as from the modulation of radiative forcing effects by regional climate variability. Further studies are needed to understand the mechanisms and causes of these differences.

Referee 2: If available, additional evidence such as historical documents and long instrumental observations are much needed to strengthen the results.

Answer: There are no longer instrumental observations because our study sites are located in remote and hardly populated regions. Gridded CRU data is not representative back in time for our sites. We provided all available evidence for historical documents discussed in detail in papers by Myglan et al., 2008; Büntgen et al., 2016; Guillet et al., 2017.

Please also note the supplement to this comment:
https://www.clim-past-discuss.net/cp-2018-70/cp-2018-70-AC2-supplement.pdf

[Figure]

**Fig. 1.** Map with the locations of the study sites (stars) and volcanic eruptions from the tropics (black dots) considered in this study (a). Annual tree-ring width index (light lines) and smoothed by 51-year Hamming window (bold lines) chronologies from northeastern Yakutia (YAK - **blue**, b) (Hughes et al., 1999; Sidorova and Naurzbaev 2002; Sidorova 2003), eastern Taimyr (TAY - **green**, c) (Naurzbaev et al., 2002), and Russian Altai (ALT - **red**, d) (Myglan et al., 2009) were constructed based on larch trees (Photos: V. Myglan – ALT, M. M. Naurzbaev – YAK, TAY).

**Fig. 1.**

[Figure]

**Fig. 2.** Normalized (z-score) individual tree-ring index chronologies (TRWi, **pink**), maximum latewood density (MXD, **black**), cell wall thickness (CWT, **green**), $\delta^{13}C$ (**red**) and $\delta^{18}O$ (**blue**) in tree-ring cellulose chronologies from YAK, TAY and ALT for the specific periods 15 years before and after the eruptions CE 535, 1257, 1640, 1815 and 1991 are presented. Vertical lines showed year of the eruptions.

**Fig. 2.**

[Figure]

**Fig. 3.** Superposed epoch analysis of $\delta^{18}O$ (a), $\delta^{13}C$ (b), CWT (c), TRW (d) and MXD (e) chronologies for each study site and for the major volcanic eruptions in CE 535, 1257, 1641, 1815 and 1991.

**Fig. 3.**

**Fig. 4.** Significant correlation coefficients between tree-ring parameters: TRW, MXD, CWT, $\delta^{13}$C and $\delta^{18}$O versus weather station data: temperature (T, red), precipitation (P, blue), vapor pressure deficit (VPD, green), and sunshine duration (S, yellow) from September of the previous year to August of the current year for three study sites were calculated. Table 2 lists stations used in the analysis.

**Fig. 4.**

[Figure]

**Fig. 5.** Response of larch trees from Siberia to the CE volcanic eruptions (Table 1) with percentile of distribution considered as very extreme (< 5th, intensive color), extreme (>5th, <10th, light color) and non-extreme (>10th, white color). July temperature changes presented as a square from **heavy blue** (cold) to **light blue** (moderate). Summer vapor pressure deficit (VPD) variabilities are

**Fig. 5.**

**Supplement:**

**Supplementary material:**

CE 536

[Figure]

[Figure]

[Figure]

CE 537

[Figure]

[Figure]

[Figure]

CE 538

[Figure]

[Figure]

[Figure]

CE 541

[Figure]

[Figure]

[Figure]

CE 542

[Figure]

[Figure]

[Figure]

CE 1258

[Figure]

[Figure]

[Figure]

CE 1259

[Figure]

[Figure]

[Figure]

CE 1453

[Figure]

[Figure]

[Figure]

CE 1458

[Figure]

[Figure]

[Figure]

CE 1601

1602

[Figure]

[Figure]

[Figure]

CE 1641

[Figure]

[Figure]

[Figure]

CE 1642

[Figure]

[Figure]

[Figure]

CE 1816

[Figure]

[Figure]

[Figure]

CE 1817

[Figure]

[Figure]

[Figure]

CE 1992

1993

---

## Referee Report (RR1)

**Review of re-revised manuscript by Churakova et al. "Siberian tree-ring and stable isotope proxies as indicators of temperature and moisture changes after major stratospheric volcanic eruptions" for Climate of the Past**

1. The new SEA fig. 3 means we can at last begin to see the behaviour of the five tree-ring parameters after eruptions. A minor comment on its interpretation: L369 "The behavior of isotope chronologies is rather more complex, with a distinct decrease in δ13C at the high-latitude sites (YAK, TAY), whereas δ18O series are impacted mainly at the high-latitude YAK and high-altitude ALT sites." The largest decrease in d18O (in terms of z-score) is in eruption+2years at the high-altitude site (ALT). So why do you highlight the high-latitude sites and not this one?

2. L459 "1816 was cold only in YAK (based on the CWT chronology), but not at the other sites". This is in agreement with Fig. 5 but is not agreement with Fig. 2 or Fig. S1 (see page 14 of the supplement), nor are Figs. 2 and S1 in agreement with each other! I already highlighted the disagreements between Fig. 2 and Fig. S1 for YAK in 1816 in my previous review (see my previous comment on L347-348) and yet the authors just responded "We carefully checked and corrected figures accordingly" and added in the comment above about 1816 YAK CWT indicating cold.

Here is the YAK panel from Fig. 2:                    And the YAK 1816 panel from Fig. S1:

[Figure]

Fig. S1 shows YAK anomalies around -2 (z-score) in 1816 (vertical dashed lines) for four out of the five parameters (TRW, MXD, C13 and CWT), only O18 (vertical dashed blue line is at zero). This is not compatible with Fig. 2 in the main text which shows only notably negative values in 1816 for YAK are for MXD (pink/purple) and the TRW (black), C13 (red) and CWT (green) do not show z-scores near -2. Which is correct, Fig. 2 or Fig. S1? And why does Fig. 5 show only notable cooling for YAK CWT in 1816 when Fig. 2 shows YAK CWT is not anomalous in 1816 (see above)?

3. There appear to be other inconsistencies between Figs. 2, 5 and S1. Take ALT 1816 for instance. Fig. 2 shows no notable excursions for any parameter in 1816 at ALT (**all** lie between +/-1 for the z-scores). Fig. 5 shows notable d13C anomaly (orange rhomb and orange circle, indicating dry (high summer VPD or low July precip). Fig. S1 (p. 14 of supplement) shows d13C anomaly in 1816 (vertical dashed red line) is almost exactly zero! So why does Fig. 5 indicate notably dry?

4. Since there have been similar inconsistencies between text and figures, or between figs. 2, 5 and S1, in previous versions of the manuscript that have not been corrected, I now have less

faith in the accuracy of what is presented and I can only encourage the authors to print out large versions of these three figures and their text and go through them site-by-site, eruption-by-eruption and parameter-by-parameter and check/correct everything. Either that or explain what the vertical dashed lines in Fig. S1 mean because the caption says they represent the anomalies in the depicted years, but they are clearly different to the anomaly timeseries in Fig. 2 for the same years.

5. In fact, I've just found another inconsistency. TAY 1817 MXD is -2.5 in Fig. S1, about -0.5 in Fig. 2 and white (indicating not notable) in Fig. 5. So is Fig. S1 wrong in this instance?

6. L461-2: You added this text in the latest version: "CE 1993 was an extremely cold year for ALT based on CWT and $\delta18O$, while also sunny, which is confirmed by local weather station data". Really? Fig. 2 for ALT shows 1993 as having CWT (green) about +1.5, and the other parameters between -1 and -1.5. So how can you conclude that high CWT implies extremely cold when CWT is positively correlated with summer temperatures (Fig. 4)?

---

## Author Response (AR2)

**Dear Prof. Luterbacher,**

**We thank you and the Reviewers for your time and valuable comments reviewing our manuscript. We believe that, thank to the constructive comments, our newly revised manuscript is significantly improved. Below, you can find the detailed point-by-point answers to the Reviewers' comments. To facilitate your assessment, we marked our responses in bold and highlight changes in the manuscript.**

**Response to the Reviewer 1**

**Reviewer:**
1) I am still a little bit worried about the robustness of these resulting conclusions based on a small network composed of only three sites.
2) In addition, chronology of different proxies and their climate response have uncertainties.
3) A related discussion on these is helpful and should be indicated in abstract and conclusion sections.

**Answer:**
**1) We agree with the Reviewer that a network with three sites is not complete, but it is the largest network existing to date. Therefore, while accepting possible limitations, we would like to underline some of the key strengths that this paper has with respect to what we knew previously:**

**i) The three sites cover a vast area of Siberia (being 1500 and 3400 km apart), including northern latitudes and high elevations. As such, they cover a rather vast territory within the boreal ecosystem. The fact that our results show that tree-ring responses to volcanic events are not consistent over time and space should help to raise questions regarding the validity of a unique volcanic forcing in climatic and dynamic global vegetation models;**

**ii) The use of a multi-proxy approach covering stratospheric volcanic eruptions over the past 1500 years remains an important effort that can only be produced at locations for which such chronologies exist. Considering the large amount of resources invested, we believe that it is still more appropriate to base our study on a set of "homogeneous sites" (boreal environment, genus *Larix*) than expanding to other regions or species.**

**iii) The multi-proxy approach proposed in a context of responses to several volcanic events is unprecedented and provides valuable results.**

**2) It is true that tree-ring proxies are characterized by "uncertainties". These are usually quantified via the assessment of the common signal and from the strength of the correlation with the climatic and/or environmental signal. In addition, there can be some additional uncertainties due to the suboptimal references of the climatic data if these are not collected at the site, as is the case in our study. This information was only partially supplied in previous versions of the submission (see results section). To make this more transparent we added the following to the text: "Mean inter-series correlation (RBAR) and EPS values of stable isotope chronologies were calculated for the period 1950-2000, for which individual trees were analyzed separately. We show the common signal with an EPS > 0.85 and series have RBAR ranging between 0.59 and 0.87. Before 1950, we used pooled material only. For all other**

tree-ring parameters, the EPS exceeds the threshold of 0.85, and RBAR values range from 0.63 to 0.94." (P. 12, L. 224-228, revised version).

Moreover, we added the following text explaining possible uncertainties in regards of weather station data (see P. 7, L. 156-161) "Due to the remote localization of our study sites, we used meteorological data from monitored weather stations located at distances ranging from 50-200 km from the sampling sites. Temperature data from these weather stations are significantly correlated ($r > 0.91$; $p < 0.05$) with gridded data (http://climexp.knmi.nl). However, poor correlation is found with precipitation data ($r < 0.45$; $p < 0.05$), most likely as a result representing local effects (Churakova (Sidorova) et al., 2016)."

Each proxy carries a specific, seasonal information, one may thus expect that the responses can be different among proxies and events, also due to delay in responses related to the location and seasonality of the eruption on the one hand and the time at which environmental changes become tangible at the study sites. Our previous work in Russian Altai (see Sidorova et al., 2012 Climate Dynamic DOI 10.1007/s00382-010-0989-6) already highlighted that different proxies can carry different signals. This has been explained by the influence of different climate parameters in temperature-limited environment, by the different seasonality and by the different response patterns to temperature and precipitation changes in the permafrost zone.

*Reference:* Sidorova OV, Saurer, M, Myglan VS, Eichler A, Schwikowski M, Kirdyanov AV, Bryukhanova MV, Gerasimova OV, Kalugin IA, Daryin AV, Siegwolf RTW A multi-proxy approach for revealing recent climatic changes in the Russian Altai Clim Dyn DOI 10.1007/s00382-010-0989-6

3) We have added more explanation on these aspects in the discussion (P. 3 L. 55-56; L.65-67, P. 12, L. 224-228, P. 7, L. 156-161). Similarly, the conclusion section was revised accordingly to the Reviewer's suggestions (P.31-32 L. 614-627).

**Reviewer:** The chosen six volcanic events do not represent the largest volcanic eruptions. Why did not they choose the top 6 in terms of stratospheric sulphur injection?

Answer:  Indeed the injection of sulfur into the stratosphere by explosive volcanic eruptions is a very important cause of significant climate variability. The selection of volcanic events was based on Table 2 in Toohey and Sigl (2017), who listed the top 20 eruptions of the past 2000 years in terms of volcanic stratospheric sulfur injection (VSSI). Our sub-criteria also include events that were well reported in tree-ring proxies (to exclude events that might have not affected the growing season and thus plant growth) and a certain temporal distribution over the last 1500 years. Therefore, we added more detailed information about the criteria used with the relative references (P. 8, L. 179- 182): "Identification of the events analyzed in this study was based on volcanic aerosols deposited in ice core records (Zielinski 1994; Robock 2000), and more precisely on Toohey and Sigl (2017) where the authors listed the top 20 eruptions from the past 2000 years in terms of volcanic stratospheric sulfur injection (VSSI) in a new ice core-based volcanic forcing reconstruction".

Our sub-criteria is based on literature review of reconstructed VSSI and events well reported in tree-ring proxies that may have had a noticeable impact on the forest ecosystems from high-latitude and high-altitude regions (Briffa et al., 1998; D'Arrigo et al., 2001; Churakova (Sidorova) et al., 2016; Büntgen et al., 2016; Gennaretti et al., 2017; Helama et al., 2018). Therefore, based on our previously published TRW and newly developed MXD, CWT,

$\delta^{13}$C and $\delta^{18}$O in tree-ring cellulose chronologies, we selected the years, characterized by strong volcanic eruptions with far-reaching climatic effect, namely the years CE 535, 540, 1257, 1640, 1815, and 1991. Therefore, to investigate climatic impacts of these eruptions in Siberian regions, we selected periods around (± 10 years): CE 525-545, 1247-1267, 1630-1650, 1805-1825, and 1950-2000, with the latter being used to calibrate tree-ring proxy versus available climate data (Table 2). (P. 9, L. 183-193)

**Response to the Reviewer 2**

**Reviewer:** Review of revised manuscript by Churakova et al. "Siberian tree-ring and stable isotope proxies as indicators of temperature and moisture changes after major stratospheric volcanic eruptions" (revised title) for Climate of the Past Discussions

As with my original review, I am supportive of this study because of its value in bringing together multiple tree-parameters and to infer multiple climatic parameters. My criticisms arose from the lack of a superposed epoch analysis (SEA), a focus on the heterogeneity of responses to different eruptions which hadn't been firmly established, and incorrect discussion of how the results from various poor quality figures had been described in the main text.

I am pleased that the authors have addressed many of my previous concerns. However there are still many problems with the manuscript. Individually, these are minor and could be easily addressed by the authors. Most are still about incorrect discussion of the results in the text that does not always match the values shown in the figures (the figures are improved, thank you). I'm annoyed that a whole section of my original review was ignored and not responded to, since some of these errors I had already highlighted and now I found them all over again!

Here is the section of my original review that was ignored (nothing mentioned in the authors' response). Of course the line numbers refer to the lines in the original 'discussions' manuscript. I repeat some of them later for the new manuscript but all the original ones should be checked anyway.

**Answer: We value reviewer's support and appreciate his careful reading. Specifically, as reported in our previous version, we indeed performed a superposed epoch analysis (SEA; See Fig. S1 in the first submission and Fig. 3 in the R1 revised version). Based on the Reviewer's comments we have tried to improve visibility of Fig. 3 in the current revised version and produced separate plots for each proxy and site separately (see revised Fig. 3, P. 20).**

**We also apologize for missing a full section in our previous revision. This was due to a mistake when merging several versions of co-authors revisions. We thank the Reviewer for his careful check and these have been now addressed in our point-by point response. The match between text and figures has been checked and inconsistencies have been solved.**

**Reviewer:** These limitations are compounded by errors in the description of the results in Fig. 4. Here are lines 439-449 with my comments in [CAPS]: "Therefore, CE 536 was extremely humid in YAK and TAY, as well as 541 and 542 [NOTHING SIGNIFICANT IN FIG 4 IN 542] in TAY and ALT. CE 1258 was dry in YAK and ALT, while drier than normal conditions occurred in 1259 for all studied sites. CE 1641 [1641 ISN'T EVEN IN FIG 4] was dry in TAY; 1642 in YAK [NO, 1643!] and ALT [NOT ACCORDING TO FIG 4]. A rather wet summer was in TAY during 1815 [THIS YEAR IS NOT IN THE FIGURE!] and 1816 years [NOTHING SIGNIFICANT HERE]. CE 1991 [NOT IN THE FIGURE!] was wet in YAK, 1992 in ALT [NO ALT DOESN'T APPEAR WET IN 1995 IN FIG 4] followed by a dry summer in 1993 [NOT ACCORDING TO FIG 4!] (Fig. 4).

**Answer: We double checked the results and figures, and revised accordingly.**

**Reviewer:** 3.3.3. Sunshine duration proxies Instrumental measurements of sunshine duration (Table 2) in YAK and ALT during the recent period showed a significant link with δ18O cellulose. Based on this we conclude that sunshine duration decreased significantly in 536, 541, 542, 1258 and 1259 in YAK, and 536 in ALT. Conversely, summer 1991 in YAK was very sunny [NOT ACCORDING TO FIG 4! BUT 1993 in ALT WAS, WHY NOT MENTION THIS?] (Fig. 4)."

**Answer: Summer 1991 in YAK was very sunny according to the weather station data (not Fig. 5). To avoid misunderstanding, we now carefully link our text with Fig. 5 (revised version), "Conversely, summer 1993 in ALT was very sunny (Fig. 5)" (see P. 26, L. 481).**

For Pinatubo, the instrumental data could be used to confirm the multi-variate explanations, e.g. do they confirm the statements you give about 1991-1993 on lines 442-443, 449 and 517?

**Answer: Yes, we used instrumental data (weather station data) to confirm our statement (P. 25, L. 461-462).**

**Reviewer:** Some description of Fig. 3 looks incorrect too, e.g. identification of months with significant correlations:
L370: "with the exception of TAY for the latter" - and CWT too? L381: "May" - looks like "April"? L391: "March" - looks like "April" for CWT?

**Answer: We corrected the sentence related to Fig. 3 as follows: "In addition, at ALT a positive relationship is observed between March precipitation and TRW ($p<0.05$) (r=0.37), MXD (r=0.32), while April precipitation is related positively with CWT (r=0.34), respectively." (P. 22, L.417-419).**

**Reviewer:** Specific comments for the revised manuscript follow.

L77: 0.5C cooling is for NH land cooling not global cooling.

**Answer:  we removed "global" from the text.**

**Reviewer:**L176: this says these events are VEI>6, but L136 says VEI exceeding 5. Table 1 has one with VEI=5 (Parker). Please be consistent, or if VEI was not the criteria for selection than don't say that it was. I understand that you've already made the selection and made the measurements, so I am not suggesting you make a different selection I'm just asking that you be consistent in stating how the selection was made. It doesn't matter if there is not a simple rule (VEI or something else). It is fine if you selected the largest plus Parker for another reason (data/samples were available, etc.) just say it.

**Answer: To avoid misunderstanding related to the VEI, we revised section 2.2 "Selection of volcanic events and larch subsamples" P. 8-9, L. 178-193: "Identification of the events analyzed in this study was based on volcanic aerosols deposited in ice core records (Zielinski 1994; Robock 2000), and more precisely on Toohey and Sigl (2017) where the authors listed the top 20 eruptions from the past 2000 years in terms of volcanic stratospheric sulfur injection (VSSI) in a new ice core-based volcanic forcing reconstruction. Our sub-criteria is based on literature review of reconstructed VSSI and events well reported in tree-ring proxies that may have had a noticeable impact on the forest ecosystems from high-latitude and high-altitude regions (Briffa et al., 1998; D'Arrigo et al., 2001; Churakova (Sidorova) et al., 2016; Büntgen et al., 2016; Gennaretti et al., 2017; Helama et al., 2018). Therefore, based on our**

previously published TRW and newly developed MXD, CWT, $\delta^{13}$C and $\delta^{18}$O in tree-ring cellulose chronologies, we selected the years, characterized by strong volcanic eruptions with far-reaching climatic effect, namely the years CE 535, 540, 1257, 1640, 1815, and 1991. Therefore, to investigate climatic impacts of these eruptions in Siberian regions, we selected periods around (± 10 years): CE 525-545, 1247-1267, 1630-1650, 1805-1825, and 1950-2000, with the latter being used to calibrate tree-ring proxy versus available climate data (Table 2)."

**Reviewer:** L228-231: "the remaining YAK sample size was too small" Should YAK be replaced by "TAY"? Otherwise I don't see how the YAK sample size causes the TAY 536 CWT value to be missing. Perhaps these sentences would be more clearly written as: "Unfortunately the remaining sample material for the CE 536 ring at TAY was insufficient to produce a clear anatomical signal. As a result, CWT is missing for CE 536 at TAY (Fig. 2)"

**Answer: We revised the sentence according to the Reviewer suggestion: "Unfortunately the remaining sample material for the CE 536 ring at TAY was insufficient to produce a clear anatomical signal. As a result, CWT is missing for CE 536 at TAY (Fig. 2)." (P. 12-13, L. 240-242).**

**Reviewer:** L240: Should it say "(g/cm3)"?

**Answer: corrected to g/cm$^3$ (P. 13, L. 251)**

**Reviewer:** Various anomalies are discussed in the results section. These do not always match the values shown in the figures, suggesting that you have not checked the values very carefully. This isn't helped by using incompatible colours in Fig. 2 and in Fig. S1 for TRW (pink vs. black) and MXD (black vs. purple).

**Answer: Colors are revised according to the reviewer's suggestions and correspond now to those in Fig. 2 and Fig. S1. Specifically, purple (Fig.2 for MXD) vs. purple (Fig.S1 for MXD), black (Fig.2 for TRW) vs. black (Fig. S1 for TRW). We have also double-checked all data and confirm that information in the Figures corresponds to data in the text.**

**Reviewer:** Some examples:
L317-318: You have added a sentence that is not supported by Fig. 2 or Fig. S1 ("Regarding CWT, a strong decrease is observed in CE 536 at YAK.").
**Answer: Regarding CWT, a strong decrease is observed in CE 536 at YAK and ALT. P. 16, L. 328;**
**Fig. 2 reference line and x-axis are corrected.**

**Reviewer:** L321-322: MXD for YAK is not -2.8 sigma in CE 537 in Fig. 2. Maybe you mean 536 for YAK and 537 for ALT?
**Answer: Corrected as follows: "…Furthermore, we revealed decreasing MXD values for ALT (-4.4 σ) in CE 537 and YAK (-2.8 σ) in CE 536." (P. 16, L. 332-333).**

**Reviewer:** L325-326: This sentence is not supported by the figures: "The δ18O chronologies show a distinct decrease one year after the eruptions for YAK -3.9 σ, in the year

of 1259, TAY -3.0σ in 537, and ALT -2.9 σ in 537 only (Fig. 2, Fig. S1)." First, 1259 and 537 are not ONE year after an eruption. Second, ALT extreme is in 536 not 537. **Answer: Thank you for this observation. We corrected as follows: "The ALT $\delta^{18}O$ chronology recorded a drastic decrease in 536 CE with (- 4.8 σ) (Fig. 2, Fig. S1). A $\delta^{18}O$ decrease for YAK was found after the CE 1257 Samalas eruption, but only in CE 1259, opposite to increased $\delta^{18}O$ values towards CE 1259 from ALT (Fig. 2). P .16, L. 335-338.**

**Reviewer:** L347-348: "No extreme anomalies are observed in CE 1816 in Siberia regardless of the site and the tree-ring parameter analyzed." Even if this is correct for your particular definition of "extreme" (lower 10th percentile) it is nevertheless worth noting that MXD at YAK does reach about -2.5 sigma in 1816, otherwise this sentence may be misinterpreted by readers. Also, I'm not sure it is correct: is -2.5 sigma really not in the lower 10th percentile (as shown in Fig. S1 for 1816 YAK). This would imply a very non-normal distribution which seems unlikely from Fig. S1 1816 YAK. Also, this does not agree with Fig. 2 of the main paper, where TRW (pink) is not less than -1 sigma, yet in Fig. S1 1816 YAK the TRW (now black) is -2.3 sigma. Also MXD is -2.5 sigma (black in Fig. 2) but -2 sigma (purple in Fig. S1). This all needs careful checking and perhaps correcting -- and perhaps Fig. 5 may also then need correcting to show cool anomalies for YAK in 1816 for either MXD or TRW depending on whether fig. 2 or fig. S1 is the correct one.

**Answer: We carefully checked and corrected figures accordingly.**

**Reviewer:** Fig. 3: it is good to now see an SEA analysis. However it's value is limited by two aspects: (1) by overlaying composites for the three sites means it is unclear what some of the values are (e.g. MXD in eruption+1 is negative for ALT and YAK but no idea what the TAY value is as there is no red bar; for eruption+2 ony the green bar is visible; etc.).

(2) no significance levels are shown -- a key advantage of SEA is that the anomaies can be tested against a null hypothesis that there is no volcanic signal. One brief mention of a statistical test is given in the text (L357-359) though it is unclear what is being averaged over. If it isn't possible to perform a statistical test of the SEA results then this should be clearly stated (e.g. because only selected years have been measured for some variables, rather than full timeseries) though I think this would only weaken the power of a test rather than prevent any test being performed.

**Answer: (1) Fig. 3 was revised according to the Reviewer's suggestions. We separated figures to avoid overlapping between plots (P. 20). (2) We applied unpaired t-test statistics to check significance between each proxy and each site (P. 16, L.322-323).**

**Reviewer:** L389-391: The results described do not seem to match those shown in Fig. 4 for d18O. There are no red bars for the months discussed at YAK and TAY. There are some for ALT but they don't appear to be in February and March.

**Answer: We removed this sentence to avoid misunderstanding because the correlations were rather low and we did not include them in the graph, but tried to discuss in the text. Therefore, we did not provide the link to the Figure.**

**Reviewer:** Fig. 5 is a very nicely designed summary. However the text describing it is not always consistent. **Answer: We revised accordingly.**

**Reviewer:** L451-452: "Therefore, CE 536 was extremely humid in YAK and TAY, as well as 541 and 542 in TAY and ALT." The only coloured symbols in Fig. 5 for CE 542 are indicators or temperature or sunshine. How can you say it was extremely humid then?

**Answer: In Fig. 5, $\delta^{13}$C values, which recorded summer vapor pressure deficit (VPD) are shown as purple circles for YAK for the years of 536, 541, for TAY for the years of 536, 537, 538 and 541 in ALT. We revised this sentence as follows: "Accordingly, the $\delta^{13}$C values showed humid summer climate conditions for YAK in 536, 541; for TAY in 536, 537, 538 and in the year of 541 for ALT. Opposite to other proxies and sites, the year of CE 537 in ALT was rather dry (Fig. 5)". P. 25, L. 468-471.**

**Reviewer:** L452-453: "CE 1258 was dry in YAK and ALT, while drier than normal conditions occurred in 1259 for all studied sites." Dry? Then why do you have purple circles in Fig. 5 indicating low vapour pressure deficits? Sure low VPD means wet?

**Answer: When VPD is increasing – drier, decreasing – wetter (by increasing air moisture).**

**We revised above-mentioned sentence according to corrected Fig.5: "Dry conditions prevailed in CE 1258 in TAY, in CE1259 in ALT, whereas wet anomalies were recorded in 1258 and 1259 in YAK" P. 25, L. 471-472.**

**Reviewer:** L453: "CE 1641 was dry in TAY; 1642 in YAK and ALT" -- but 1641 is not shown in Fig. 5. Why not? And 1642 has no coloured symbols to mark moisture at all.

**Answer: L453 revised to: "No anomalies were recorded for the CE 1642 event, irrespective of the sites." P. 25, L. 472-473.**

**Reviewer:** I just realised that many of these were in my original review and have been ignored by the authors. They have not included any of these comments in their "author response" and now I have spent my time reporting the same errors for a second time!]

**Answer: We are thankful for the Reviewer's time and appreciate the very valuable comments. We apologize for missing a full section in our previous revision. This was due to a mistake when merging several versions of co-authors revisions. We now considered all of the Reviewer's suggestions and comments, and corrected them accordingly.**

**Reviewer:** Fig. 2: I would recommend adding a vertical black line at CE 540, as currently this eruption is not marked in Fig. 2 but all the others are.

**Answer: We added a vertical black line at CE 540 as requested (revised Fig. 2, P. 18).**

**Reviewer:** SM: Much clearer now. The figure caption has disappeared, however. Despite the authors' claim, they still don't show superposed epoch analyses since all years are shown separately rather than superposed to obtain a composite. But an SEA has been included in the main figure 3.

**Answer:** We added a figure caption to Fig. S1 and revised Fig. 3 () according to the Reviewer's suggestions.

Fig. S1. Probability density function (Pdf) computed for each of the tree-ring parameter for northeastern Yakutia (YAK) (left panel), eastern Taimyr (TAY) (middle panel) and Russian Altai (ALT) (right panel), respectively. Tree-ring parameters (TRWi - black, MXD – violet, CWT – green, $\delta^{18}$O - blue and $\delta^{13}$C - red) in bold lines represent the probability density function. Dotted lines represent the anomalies (z-score, standard deviation) induced by the volcanic years: CE 536, 537, 538, 541, 542, 1258, 1259, 1453, 1458, 1601, 1602, 1642, 1643, 1816, 1817, 1992, 1993.

---

## Author Response (AR3)

Dear Prof. Luterbacher,

We would like to thank you and the referees for the comments and suggestions made on the manuscript. The manuscript has been modified quite a lot since the first submission and it is true that several figures were no longer consistent with each other. Therefore, and to be consistent, we revised Fig. 2, Fig. 5, designed a new Fig. 3 and double-checked Fig. S1.

Initially we measured and analyzed all the proxies over longer periods: CE 520-560, 1242-1286, 1625-1660, 1790-1835, 1950-2000 and all calculations performed were based on these time spans. However, in our first submission to the journal, we tried to simplify Fig. 2 and therefore presented only 10 years before and 10 years after the eruptions. However, in the supplementary material (Fig. S1), we still considered the full available time series over the periods (n=219). Therefore, there are differences between Fig 2 and Supplementary material Fig. S1. This discrepancy also led to value discrepancies in the text. Then in the process of controlling all data and calculation procedures, we also found a mistake in our script, which introduced a shift in 1816 and 1993.

We are happy that, thanks to the persistent and constructive comments of the reviewer, our latest thorough revision of the manuscript and figures has allowed us to removed all incongruences. We apologize for the inconveniences that were caused by these Figure discrepancies. Below, you can find the detailed point-by-point answers to the Reviewer's comments. To facilitate your assessment, we marked our responses in bold and highlighted the changes in the manuscript. We are providing both clean and track-mode changes versions.

**Response to the Reviewer**

**Reviewer:**
1.      The new SEA fig. 3 means we can at last begin to see the behaviour of the five tree-ring parameters after eruptions.  A minor comment on its interpretation: L369 "The behavior of isotope chronologies is rather more complex, with a distinct decrease in δ13C at the high-latitude sites (YAK, TAY), whereas δ18O series are impacted mainly at the high-latitude YAK and high-altitude ALT sites." The largest decrease in d18O (in terms of z-score) is in eruption+2years at the high---altitude site (ALT). So why do you highlight the high---latitude sites and not this one?

**Answer 1: We re-plotted Fig. 3 by selecting 15 years before and 20 years after the eruptions (CE 535, 540, 1257, 1640, 1815) to better visualize deviation and duration of negative effects after the volcanic eruptions on Siberian trees (see below). For CE 1991, we considered 15 years prior to the event as well. After the event, the shortest series only has 9 years, however. We agree, that z-scores of $\delta^{18}$O values from ALT in the first year after the eruptions decreased strongly compared to YAK and TAY sites. However, in terms of duration negative values in tree-ring parameters from the high-latitude sites (YAK and TAY) prevailed compared to high-altitude ALT site (revised Fig. 3 below).**

[Figure]

$\delta^{18}O$

$\delta^{13}C$

**At the ALT site, $\delta^{18}O$ become positive 4 years after the eruptions, while at the TAY site, $\delta^{18}O$ values remained negative for 12 years with a response delay of 2 years after the eruption. At the YAK site, positive $\delta^{18}O$ value was revealed on the third year after the eruptions, followed by 7 years of negative values afterwards (in total longer, compared to ALT). For clarification we rephrased the sentences as follows:**

**"The $\delta^{18}O$ isotope chronologies (z-score) show a distinct decrease the year after the eruptions. At ALT, however, the duration of negative anomalies were shorter (5 years) than at the high-latitude TAY (12 years) and YAK (9 years) sites. At the YAK site, two negative years followed the events, intermitted with one positive value, to remain negative during the following 7 years. The duration of negative anomalies recorded in $\delta^{13}C$ values (z-score) lasts also longer at the high-latitude YAK site - 10 years after the eruptions and 13 years at TAY compared to 7 years at ALT (Fig. 3)." P.20, L. 373-379.**

**Reviewer:**

2.        L459 "1816 was cold only in YAK (based on the CWT chronology), but not at the other sites". This is in agreement with Fig. 5 but is not agreement with Fig. 2 or Fig. S1 (see page 14 of the supplement), nor are Figs. 2 and S1 in agreement with each other! I already highlighted the disagreements between Fig. 2 and Fig. S1 for YAK in 1816 in my previous review (see my previous comment on L347---348) and yet the authors just responded "We carefully checked and corrected figures accordingly" and added in the comment above about 1816 YAK CWT indicating cold.

**Answer 2: There was indeed an error in the calculation of the pdfs for the Fig. S1 in the previous submission. We apologize for the inconvenience and thank the referee for spotting our mistake. We**

**double-checked all the values and figures and revised them. The revised figures are shown below (the old figures have been added as comparison):**

**REVISED Fig. 2:**                                    **OLD Fig. 2:**

[Figure]

From revised Fig. 2: Normalized (z-score) individual tree-ring index chronologies (TRW, black), maximum latewood density (MXD, **purple**), cell wall thickness (CWT, **green**), $\delta^{13}$C (**red**) and $\delta^{18}$O (**blue**) in tree-ring cellulose chronologies from northeastern Yakutia (YAK) for the specific period CE 1790-1835 before and after the eruption (CE 1815) are presented. Vertical lines show year of the eruptions.

**REVISED Fig. S1     YAK 1815**                    **OLD: Fig. S1     YAK 1816**

[Figure]

From Fig. S1. Probability density function (Pdf) computed for each of the tree-ring parameter for northeastern Yakutia. Tree-ring parameters (TRWi - black, MXD – purple, CWT – green, $\delta^{18}$O - blue and $\delta^{13}$C - red) in bold lines represent the probability density function. Dotted lines represent the anomalies (z-score) observed for the first and second years following the Tambora volcanic eruption (CE 1815) for each tree-ring parameter.

**Reviewer:** Fig. S1 shows YAK anomalies around ---2 (z-score) in 1816 (vertical dashed lines) for four out of the five parameters (TRW, MXD, C13 and CWT), only O18 (vertical dashed blue line is at zero). This is not compatible with Fig. 2 in the main text which shows only notably negative values in 1816 for YAK are for MXD (pink/purple) and the TRW (black), C13 (red) and CWT (green) do not show z-scores near -2. Which is correct, Fig. 2 or Fig. S1?  And why does Fig. 5 show only notable cooling for YAK CWT in 1816 when Fig. 2 shows YAK CWT is not anomalous in 1816 (see above)?

**Answer: In Fig. 2 we changed the pink line to purple and extended the periods to match those of Fig.S1. The data have been carefully double-checked and differences corrected. In Fig. 5 it is correct that YAK shows an anomalous 1816 –MXD (and not CWT).**

**Reviewer:**

3.      There appear to be other inconsistencies between Figs. 2, 5 and S1.  Take ALT 1816 for instance. Fig. 2 shows no notable excursions for any parameter in 1816 at ALT (all lie between +/---1 for the z---scores).  Fig. 5 shows notable d13C anomaly (orange rhomb and orange circle, indicating dry (high summer VPD or low July precip).  Fig. S1 (p. 14 of supplement) shows d13C anomaly in 1816 (vertical dashed red line) is almost exactly zero! So why does Fig. 5 indicate notably dry?

**Answer 3: Correct, there is no notable excursion for any parameter in 1816 at ALT. We corrected this mistake. The colors of Fig.5 have been revised, now using dark blue, light blue and white. The $\delta^{13}$C anomaly for the humid year 1817 has been corrected to light blue.**

**Reviewer:**

4.      Since there have been similar inconsistencies between text and figures, or between figs. 2, 5 and S1, in previous versions of the manuscript that have not been corrected, I now have less faith in the accuracy of what is presented and I can only encourage the authors to print out large versions of these three figures and their text and go through them site-by-site, eruption-by-eruption and parameter-by-parameter and check/correct everything.  Either that or explain what the vertical dashed lines in Fig. S1 mean because the caption says they represent the anomalies in the depicted years, but they are clearly different to the anomaly timeseries in Fig. 2 for the same years.

**Answer 4: We now have revisited all figures and the text as suggested by the referee. Specifically, we revised and corrected Figures 2, 5, S1 and double-checked all values mentioned in the text.**

**The dotted, vertical lines represent the anomalies (z-score) observed for the first and second years following the CE 535, 540, 1257, 1640, 1815 and 1991 volcanic eruptions for each tree-ring parameter. We agree with the referee that there were inconsistencies between Fig. 2 and Fig S1. These have now been corrected and removed.**

**Reviewer:**

5.      In fact, I've just found another inconsistency.  TAY 1817 MXD is -2.5 in Fig. S1, about -0.5 in Fig. 2 and white (indicating not notable) in Fig. 5.  So is Fig. S1 wrong in this instance?

**Answer 5: We thank the referee for the careful review. However, no anomalies occurred at TAY in 1816 or 1817 (Fig. 2, 5). MXD 1816 is correct for YAK.**

**Reviewer:**

6.      L461-2: You added this text in the latest version: "CE 1993 was an extremely cold year for ALT based on CWT and δ18O, while also sunny, which is confirmed by local weather station data".  Really? Fig. 2 for ALT shows 1993 as having CWT (green) about +1.5, and the other parameters between -1 and -1.5.  So how can you conclude that high CWT implies extremely cold when CWT is positively correlated with summer temperatures (Fig. 4)?

Answer 6: **Correct, CWT is positively correlated with summer temperatures. CE 1993 was not an extremely cold year at ALT based on CWT and δ$^{18}$O. Negative value in CWT was recorded at YAK site, not ALT. We corrected the sentence as follows:**

[revised manuscript text omitted]